# The missing linker between SUN5 and PMFBP1 in sperm head-tail coupling apparatus

Ying Zhang [1,5], Chao Liu[2,3,5], Bingbing Wu[3,4], Liansheng Li[1], Wei Li [2,3,4✉] & Li Yuan [1✉]

The sperm head-to-tail coupling apparatus (HTCA) ensures sperm head-tail integrity while defective HTCA causes acephalic spermatozoa, rendering males infertile. Here, we show that CENTLEIN is indispensable for HTCA integrity and function, and that inactivation of CEN-TLEIN in mice leads to sperm decapitation and male sterility. We demonstrate that CENTLEIN directly interacts with both SUN5 and PMFBP1, two proteins localized in the HTCA and related with acephalic spermatozoa syndrome. We find that the absence of *Centlein* sets SUN5 and PMFBP1 apart, the former close to the sperm head and the latter in the decapitated tail. We show that lack of *Sun5* results in CENTLEIN and PMFBP1 left in the decapitated tail, while disruption of *Pmfbp1* results in SUN5 and CENTLEIN left on the detached sperm head. These results demonstrate that CENTLEIN cooperating with SUN5 and PMFBP1 participates in the HTCA assembly and integration of sperm head to the tail, indicating that impairments of *CENTLEIN* might be associated with acephalic spermatozoa syndrome in humans.

[1] Savaid Medical School, University of Chinese Academy of Sciences, Beijing, P.R. China. [2] Fertility Preservation Lab, Reproductive Medicine Center, Guangdong Second Provincial General Hospital, Guangzhou, P.R. China. [3] State Key Laboratory of Stem Cell and Reproductive Biology, Institute of Zoology, Stem Cell and Regenerative Medicine Innovation Institute, Chinese Academy of Sciences, Beijing, P.R. China. [4] College of Life Sciences, University of Chinese Academy of Sciences, Beijing, P.R. China. [5]These authors contributed equally: Ying Zhang, Chao Liu. ✉email: leways@ioz.ac.cn; yuanli@ucas.ac.cn

The structural and functional integrity of human spermatozoa is essential for male fertility, and any defects of the spermatozoa might be associated with severe male infertility. Many teratozoospermia have been reported to cause male infertility[1]. One of the most extreme forms of teratozoospermia is acephalic spermatozoa syndrome, which is characterized by decapitated flagella, very few intact spermatozoa, and tailless sperm heads in the semen, and it finally leads to severe male infertility[2,3]. In the past decades, studies show the acephalic spermatozoa syndrome with familial clustering, suggesting that this is a syndrome with a specific genetic origin[4–8]. Many mouse models display acephalic spermatozoa phenotype to some extent[9–19], among which Spata6-null mice are the first to display uniformly 100% acephalic spermatozoa[16]. As SPATA6 could interact with myosin light-chain polypeptide 6 on the manchettes of elongating spermatids, it might be involved in myosin-based microfilament transport for the assembly of the segmented columns (Scs) and capitulum (Cp) during sperm head–tail coupling apparatus (HTCA) formation[16]. Recently, new biological techniques together with pedigree analysis enable us identify and verify two acephalic spermatozoa syndrome-causing genes[19,20]. Mutations in Sad1 and UNC84 domain–containing 5 (SUN5) and Polyamine modulated factor 1 binding protein 1 (PMFBP1) to date account for ~70% of all cases with acephalic spermatozoa syndrome[19,20]. SUN5 is a transmembrane protein located in the nuclear envelope (NE), which could interact with the coupling apparatus-related protein DnaJ heat shock protein family (Hsp40) member B13 (DNAJB13) to facilitate SUN5 protein folding in order to ensure the interaction between the implantation fossa and an unknown protein in the HTCA[21]. PMFBP1 was first identified as a polyamine modulating factor 1 binding protein, which could enhance the catabolism and recycling of polyamines[22]. During spermatogenesis, PMFBP1 is specifically expressed in adult testes and predominantly located in the HTCA[19,23]. Although SUN5 and PMFBP1 do not physically interact with each other, both of them localize to the HTCA[19]. Together with SPATA6, they form a sandwich-like structure in HTCA: SUN5 acts as the root that connects the HTCA to the sperm NE, while PMFBP1 is located in the middle region between SUN5 and SPATA6[19,24]. Although their relative position in HTCA is well established, the linker(s) between them is still unknown.

Spermatogenesis is the process of haploid male gamete production with successive cellular differentiation[25,26]. During spermatogenesis, germ cells undergo meiosis to ensure haploidization of the genome and genetic diversity[27,28], and the haploid germ cells subsequently undergo a dramatic morphological change and nuclear chromatin re-organization to form spermatozoon, during this process, the cytoplasm needs to be removed, and forming two specific structures termed as acrosome and flagellum[28,29]. The HTCA in the neck region (also known as the connecting piece) of the mammalian spermatozoon is indispensable for the integration of the sperm head and tail and was described extensively in the late 1960s and early 1970s[30–33]. Ultrastructural studies reveal that this complex structure has a dense, convex articular region called the Cp that conforms to the concavity of the basal plate (Bp), an electron-dense structure, lining the implantation fossa of the nucleus. The Bp and the Cp are interconnected by fibrous structures that might mediate sperm head-to-tail anchorage[30]. Extending backward from the Cp are nine cylindrically Scs with periodic densities[31]. At the caudal end, each Sc is continuous with one of the nine outer dense fibers (ODFs) that associate with peripheral microtubular doublets of the growing axoneme[24]. Beneath the articular surface of the Cp is a cylindrical niche that encloses the proximal centriole (Pc) that is oriented transversely[30], in the same plane as the flattening of the

head. There is also a distal centriole (Dc) at the base of the axoneme oriented approximately at a right angle to the Pc[30,34–36]. Although these structures have been well described, the molecular composition and assembly properties are still unknown. However, according to these previously described HTCA assembly processes, the centriole-related proteins are expected to be involved in HTCA assembly.

To identify the linker(s) between SUN5 and PMFBP1, we conducted a small-scale screening of the centrosome-related proteins by co-immunoprecipitation with SUN5 and identified CENTLEIN as a direct linker between SUN5 and PMFBP1. Knockout of Centlein in mice led to the production of acephalic spermatozoa and male sterility. Ultrastructural analyses revealed that depletion of CENTLEIN resulted in the disruption of HTCA assembly at early steps of the round spermatids with complete lack of Scs and Cp. The knockout of Centlein resulted in sperm head and tail broken between SUN5 and PMFBP1. And the knockout of Sun5 resulted in CENTLEIN and PMFBP1 left on the tip of the decapitated tail, while the knockout of Pmfbp1 resulted in SUN5 and CENTLEIN left on broken sperm head only. All these results suggest that CENTLEIN acting as a bona fide linker between SUN5 and PMFBP1 participates in HTCA assembly and integrates sperm head to the tail.

## Results

**CENTLEIN interacts with both SUN5 and PMFBP1**. To identify the potential linker between SUN5 and PMFBP1, eight testis that predominantly expressed centrosome-related proteins were selected as the first batch of screening for their interaction with SUN5. To this end, we transfected HEK293T cells with a green fluorescent protein (GFP)-tagged plasmid encoding a centrosome protein (Supplementary Fig. 1) and FLAG-tagged SUN5 and then performed anti-GFP-immunoprecipitations followed by western blotting (Fig. 1a). Only CENTLEIN was present in FLAG-SUN5 immunoprecipitate(s) (Fig. 1a, b). In the reverse direction, SUN5 could be detected in the GFP-tagged CENTLEIN immunoprecipitate(s), but not in the control sample (Fig. 1c).

We then tested whether CENTLEIN could interact with PMFBP1 and/or SPATA6. Although SPATA6 could not bind to CENTLEIN (see below), epitope-tagged Centlein and Pmfbp1 expressed in HEK293T cells were able to interact with each other in reciprocal immunoprecipitation experiments (Fig. 1d–g). Taken together, CENTLEIN acts as the missing linker by mediating an interaction between PMFBP1 and SUN5, suggesting its involvement in maintenance of HTCA integrity.

**_Centlein_ knockout leads to acephalic spermatozoa and male sterility**. To determine the physiological role of CENTLEIN, we generated Centlein-knockout founder mice by applying the CRISPR-Cas9 system that targeted exon 1 and exon 23 of the Centlein gene (Supplementary Fig. 2a, b). Eight founders were obtained, and one heterozygous mutated mouse with 216088 base pair deletion from exon 1 to exon 23 of Centlein (Supplementary Fig. 2a, b) was further bred to wild-type (WT) mice; the resulting heterozygotes were interbred to obtain homozygous Centlein$^{-/-}$ mice (Supplementary Fig. 2c), which were genotyped by genomic DNA PCR (Supplementary Fig. 2c). The CENTLEIN protein was completely absent in the Centlein$^{-/-}$ testis compared with the Centlein$^{+/-}$ and Centlein$^{+/+}$ testes (Fig. 2a), indicating that homozygous mice were Centlein-null. Inbreeding of heterozygous littermates gave WT (+/+), heterozygous (+/−), and Centlein knockout (−/−) offspring at the expected Mendelian ratio (+/+: +/−:−/− = 49:105:48), indicating that the Centlein$^{-/-}$ mouse was viable.

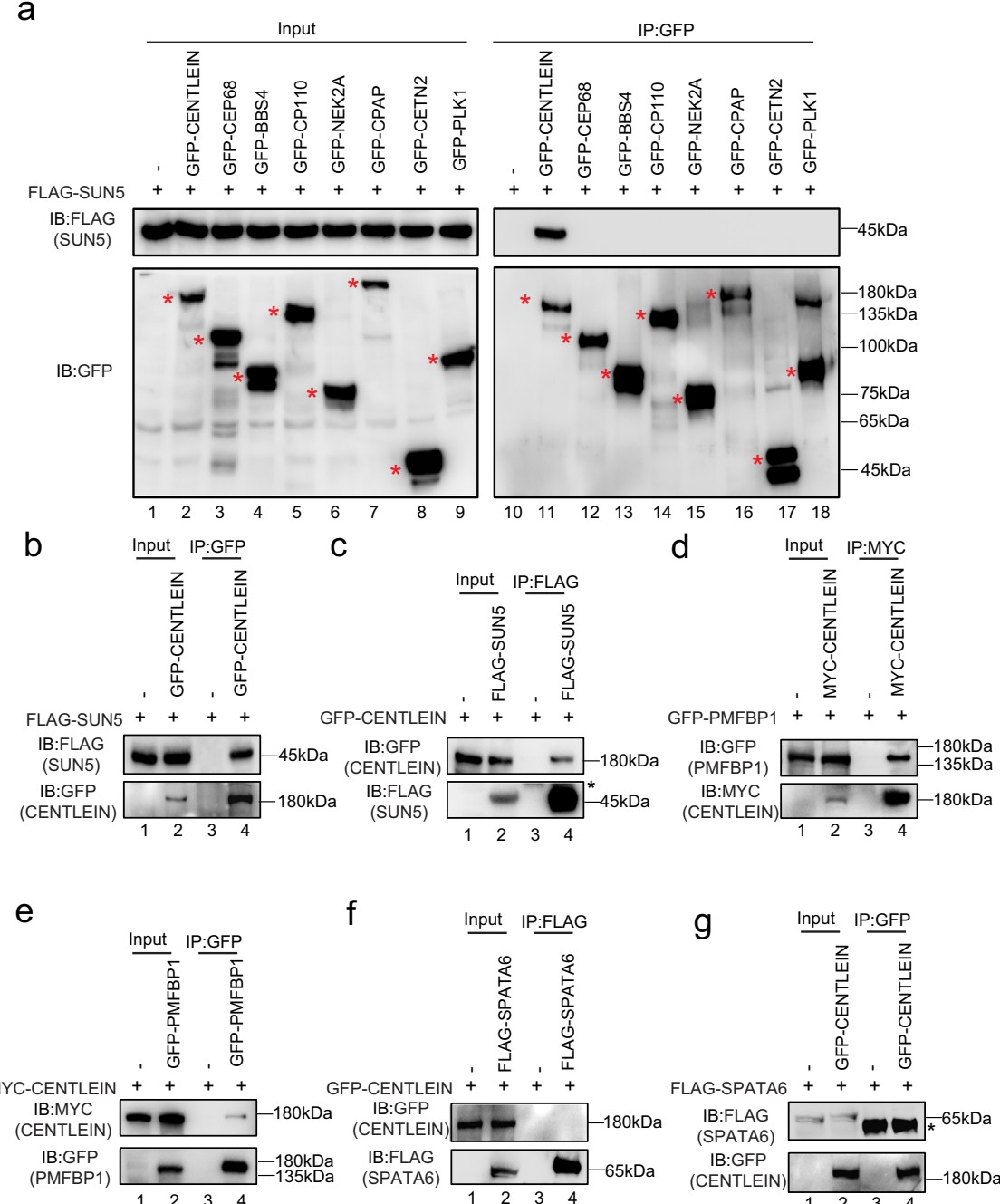

**Fig. 1 CENTLEIN interacts with both SUN5 and PMFBP1. a** A candidate-based approach by identification of SUN5-binding centrosomal proteins. FLAG-SUN5 and one of the GFP-tagged plasmid, including the empty vector, GFP-CENTLEIN, GFP-CEP68, GFP-BBS4, GFP-CP110, GFP-NEK2A, GFP-CPAP, GFP-CETN2, and GFP-PLK1, were co-transfected into HEK293T cells. Twenty-four hours after transfection, cells were collected for immunoprecipitation (IP) with anti-GFP antibody and analyzed with FLAG and GFP antibodies, respectively. Red asterisks indicate GFP-tagged proteins. **b**, **c** CENTLEIN could bind with SUN5. Empty vector, GFP-CENTLEIN and FLAG-SUN5 were co-transfected into HEK293T cells. Twenty-four hours after transfection, cells were collected for immunoprecipitation (IP) with anti-GFP antibody (**b**) or anti-FLAG antibody (**c**) and analyzed with FLAG and GFP antibodies, respectively. Asterisk indicates IgG heavy chains. **d**, **e** CENTLEIN could interact with PMFBP1. Empty vector, MYC-CENTLEIN, and GFP-PMFBP1 were co-transfected into HEK293T cells. Twenty-four hours after transfection, cells were collected for immunoprecipitation (IP) with anti-MYC antibody (**d**) or anti-GFP antibody (e) and analyzed with MYC and GFP antibodies, respectively. **f**, **g** CENTLEIN could not interact with SPATA6. Empty vector, GFP-CENTLEIN, and FLAG-SPATA6 were co-transfected into HEK293T cells. Twenty-four hours after transfection, cells were collected for immunoprecipitation (IP) with anti-FLAG antibody (**f**) or anti-GFP antibody (**g**) and analyzed with FLAG and GFP antibodies, respectively. Asterisk indicates IgG heavy chains. The experiment was repeated three times independently with similar results (**a**–**g**).

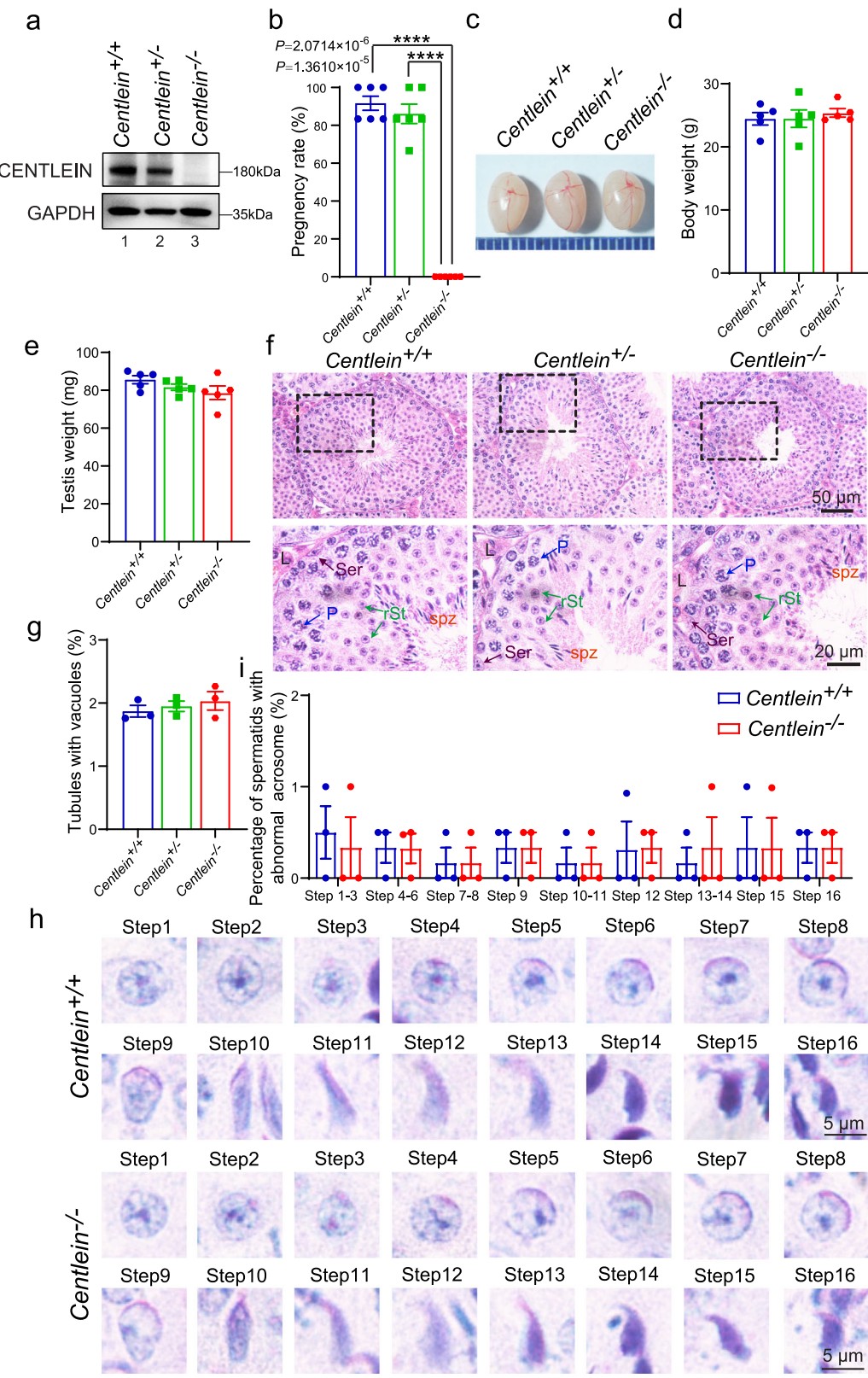

*Centlein$^{-/-}$* mice could grow to adulthood and were apparently healthy (see below), but mating outcomes were not normal (Fig. 2b). When we conducted cross-breeding experiments using *Centlein$^{-/-}$* male mice, we found that *Centlein$^{-/-}$* male mice failed to produce any offspring (Fig. 2b). Of note, *Centlein$^{-/-}$* male mice exhibited normal libido when presented with a female mouse in mating cages, and evidence of ejaculation

in the form of vaginal plugs was usually found within a few hours. Thus, the *Centlein$^{-/-}$* male mice were infertile, and disruption of *Centlein* may influence the spermatogenesis.

To further explore the cause of the male infertility, we first examined the *Centlein$^{-/-}$* testis at gross and histological levels. We found no significant differences in the testis size and weight among *Centlein$^{-/-}$*, *Centlein$^{+/-}$*, and *Centlein$^{+/+}$* mice

**Fig. 2 The disruption of *Centlein* in mice leads to male infertility. a** The CENTLEIN protein was completely absent in the *Centlein*⁻/⁻ testis. Immunoblotting of CENTLEIN was performed in the *Centlein*⁺/⁺, *Centlein*⁺/⁻, and *Centlein*⁻/⁻ testes. GAPDH served as a loading control. Biologically independent mice were examined in three separate experiments with similar results. **b** *Centlein*-deficient male mice were completely infertile. The fertility assessment experiments were performed in *Centlein*⁺/⁺, *Centlein*⁺/⁻, and *Centlein*⁻/⁻ male mice ($n = 6$ independent experiments). Data are presented as mean ± SEM. A two-tailed Student's *t* test was performed, ****$P < 0.0001$. **c** The size of the testes was not altered in the *Centlein*⁺/⁺, *Centlein*⁺/⁻, and *Centlein*⁻/⁻ mice. **d** Quantification ratio of body weight in *Centlein*⁺/⁺, *Centlein*⁺/⁻, and *Centlein*⁻/⁻ male mice ($n = 5$ independent experiments). Data are presented as mean ± SEM. **e** Quantification ratio of testis weight in *Centlein*⁺/⁺, *Centlein*⁺/⁻, and *Centlein*⁻/⁻ male mice ($n = 5$ independent experiments). Data are presented as mean ± SEM. **f** The histomorphology of *Centlein*-deficient seminiferous tubules was similar to the control groups as shown by H&E staining. L: Leydig cells, Ser (brown): Sertoli cells, P (blue): pachytene spermatocytes, rSt (green): round spermatid, spz (red): spermatozoa. **g** Quantification ratio of seminiferous tubules with vacuoles in the *Centlein*⁺/⁺, *Centlein*⁺/⁻, and *Centlein*⁻/⁻ testes ($n = 3$ independent experiments). Data are presented as mean ± SEM. **h** Acrosome and nucleus morphology in different steps of spermatid development was normal in *Centlein*-deficient mice. The periodic acid-Schiff (PAS) and hematoxylin staining was performed in *Centlein*⁺/⁺ and *Centlein*⁻/⁻ mouse. **i** Quantification ratio of spermatids with abnormal acrosome in the *Centlein*⁺/⁺ and *Centlein*⁻/⁻ testes ($n = 3$ independent experiments). Data are presented as mean ± SEM. Source data are provided as a Source data file. Blue dots indicate *Centlein*⁺/⁺ mice, green dots indicate *Centlein*⁺/⁻ mice, and red dots indicate *Centlein*⁻/⁻ mice.

(Fig. 2c–e). The hematoxylin and eosin (H&E) staining revealed that the seminiferous tubules of *Centlein*⁻/⁻ mice were also similar to the control groups, and all of the components in the seminiferous epithelium could be observed in the *Centlein*⁻/⁻ testis (Fig. 2f, g). Further periodic acid-Schiff (PAS)–hematoxylin staining showed that the acrosome and nucleus morphology in *Centlein*⁻/⁻ spermatids was normal (Fig. 2h, i), indicating that disruption of *Centlein* has little effect on meiosis and acrosome biogenesis.

Next, we examined the spermatozoa in the caudal epididymis and found that the sperm count in the *Centlein*⁻/⁻ caudal epididymis was significantly decreased compared with the control groups (Fig. 3a, b). Although the spermatozoa in the *Centlein*⁻/⁻ caudal epididymis appeared to be stained less with hematoxylin compared with those in the *Centlein*⁺/⁻ and *Centlein*⁺/⁺ caudal epididymis (Fig. 3a), eosin staining in *Centlein*⁻/⁻ caudal epididymis was indistinguishable from the control groups, suggesting that sperm heads in the *Centlein*⁻/⁻ epididymis are much less abundant. Further morphological evaluation revealed that 94.21 ± 0.31% *Centlein*-null spermatozoa in the caudal epididymis were negative for both peanut agglutinin (PNA) and 4′,6-diamidino-2-phenylindole (DAPI) staining (Fig. 3c, d). Transmission electron microscopic (TEM) analysis of the *Centlein*⁻/⁻ caudal epididymis also showed that the *Centlein*-null spermatozoa had no sperm head and contained a residual droplet of cytoplasm at the top of the flagellum with misarranged mitochondria inside (Fig. 3e). In addition, the axoneme and ODFs of decapitated tails in *Centlein*⁻/⁻ mice were also perturbed (Fig. 3e–i). Therefore, disruption of *Centlein* leads to the production of acephalic spermatozoa in mice, which may be responsible for the *Centlein*⁻/⁻ male infertility.

**The disruption of *Centlein* impaired head-to-tail anchorage of the spermatids.** To delineate how the disruption of CENTLEIN causes acephalic spermatozoa, we first examined wherein the flagellum detached from sperm head in *Centlein*⁻/⁻ mice. We detected the proportion of decapitated tails in the *Centlein*⁻/⁻ corpus and caput of the epididymis and found that they were similar to those in the *Centlein*⁻/⁻ caudal epididymis (Figs. 3d and 4a). Thus, the detachment of the sperm head and tail in *Centlein*⁻/⁻ mice may occur within the seminiferous tubules or entrance into the caput of the epididymis. Then we carefully examined the stages of spermiogenesis by PAS–hematoxylin staining in the *Centlein*⁻/⁻ and *Centlein*⁺/⁺ testes. Although the acrosome biogenesis and the process of the sperm head shaping showed normal in *Centlein*⁻/⁻ mice (Figs. 2h, i and 4b, c), the orientation of the sperm head at stages V–VIII was toward the lumen of the seminiferous tubules, but not toward the basement membrane (Fig. 4b, d, e), which might be caused by the sperm

head detachment from flagellum during spermiogenesis (Fig. 4b, d, e). Furthermore, the mature sperm head could still be observed at stages IX–X in the *Centlein*⁻/⁻ testes, while mature spermatozoa were released into the lumen of the seminiferous tubule at stage VIII in the *Centlein*⁺/⁺ testes (Fig. 4b), suggesting that the sperm head and tail might break apart during spermiation in the *Centlein*⁻/⁻ testes.

Next, we examined the HTAC in *Centlein*⁻/⁻ and *Centlein*⁺/⁺ spermatids by TEM. In *Centlein*⁺/⁺ step 7–9 spermatids, the ultrastructure of the HTCA was fully assembled and consisted of a well-defined Sc, Cp, Pc, Dc, and Bp (Fig. 4f). Having spermatids elongated, the *Centlein*⁺/⁺ spermatid coupling apparatus together with the flagellum was always attached to the NE (Fig. 4f). In stark contrast, two types of HTCA ultrastructures were detected in *Centlein*⁻/⁻ mice: Type I showed that the HTCA could not be assembled at early steps of the spermatids (steps 7–9) and the destroyed coupling apparatus scattered in the cytosol of the elongated spermatid (Fig. 4f and Supplementary Fig. 3); Type II displayed that anomalous Sc and Bp were present but far from its native implantation site in the elongated spermatids (Fig. 4f and Supplementary Fig. 3). Because of the ODFs descending from the Scs, we often observed severe flagellar defects in *Centlein*-null spermatozoa (Fig. 3e–i). These results suggest that CENTLEIN might play two functional roles in HTCA: first, CENTLEIN might work as a bona fide centrosomal component initiating assembly of the ultrastructural components of the HTAC; second, CENTLEIN might be required for the tight attachment of the coupling apparatus to the caudal portion of the sperm head. The lack of CENTLEIN leads to the destroyed coupling apparatus detached from the sperm nucleus during spermiogenesis, causing the production of acephalic spermatozoa.

**CENTLEIN localizes at the HTCA.** CENTLEIN is a centriolar protein, and we demonstrate that CENTLEIN could mediate an interaction between C-Nap1 and Cep68 to maintain centrosome cohesion[37]. To investigate the physiological function of CENTLEIN during spermatogenesis, we examined its expression and found that CENTLEIN was predominantly expressed in the testis (Fig. 5a), albeit with a weak expression in the ovary (Fig. 5a). The CENTLEIN was detectable in the testes at postnatal day (P) 14 (P14), and the level kept increasing from P28 onward, with the highest levels detected in the adult testes (Fig. 5b), indicating that CENTLEIN was expressed in late pachytene spermatocytes and spermatids. To further confirm it, we characterized its precise localization during spermiogenesis using super-resolution microscopy. In early round spermatids, CENTLEIN could partially co-localize with centrosomal protein CEP135[38] nearby the nucleus (Fig. 5c). During the elongation and differentiation of the spermatid, CENTLEIN, together with CEP135, localized to the

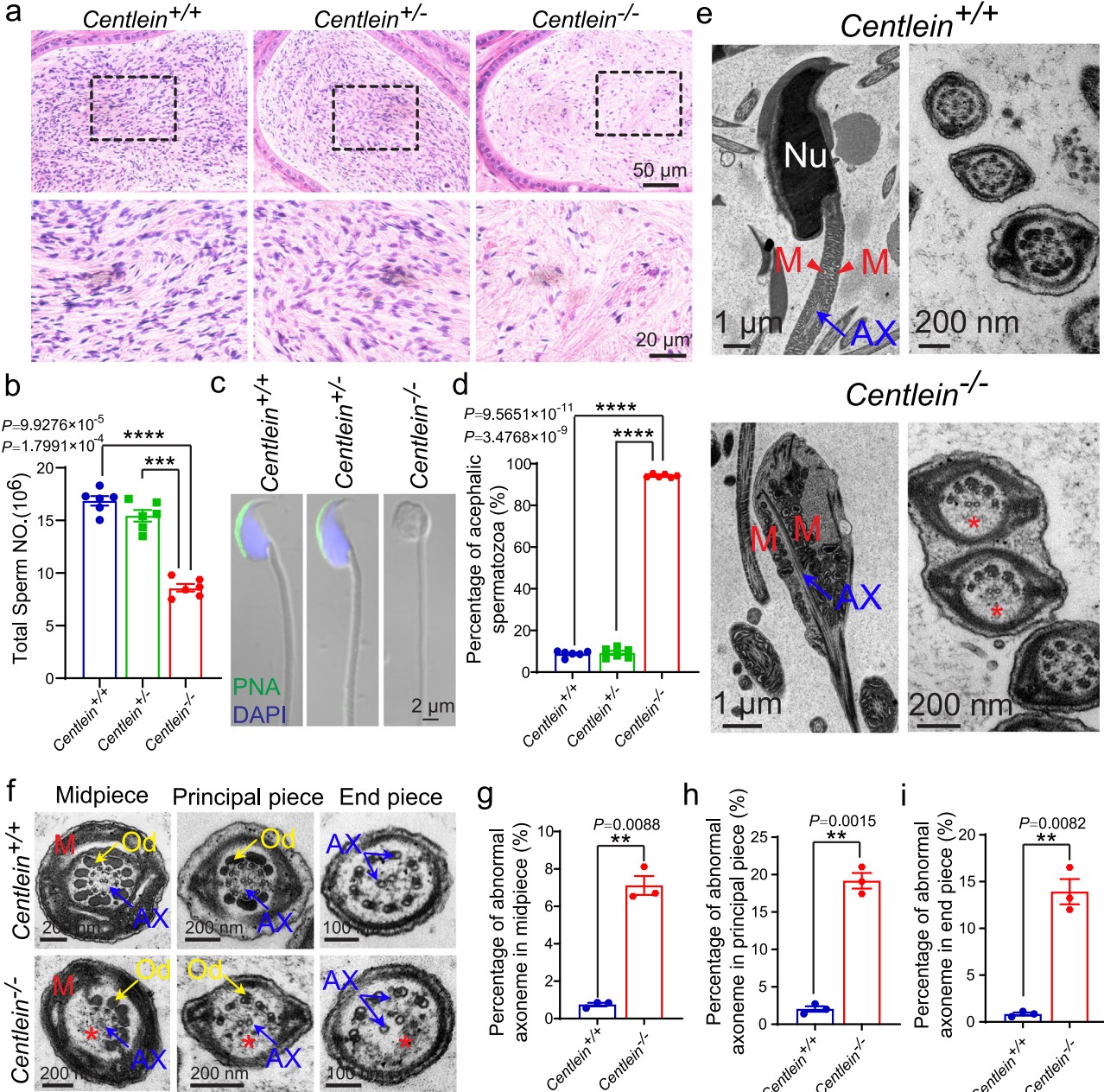

**Fig. 3 Ablation of *Centlein* leads to acephalic spermatozoa. a** Fewer sperm heads are present in *Centlein*$^{-/-}$ epididymis. The H&E staining of the caudal epididymis of *Centlein*$^{+/+}$, *Centlein*$^{+/-}$, and *Centlein*$^{-/-}$ mice are shown. The spermatozoa within the *Centlein*-deficient caudal epididymis appear to be stained less with hematoxylin compared with those in the *Centlein*$^{+/+}$ and *Centlein*$^{+/-}$ caudal epididymis. Biologically independent mice for each genotype were examined in three separate experiments with similar results. **b** The sperm counts in the caudal epididymis of *Centlein*$^{-/-}$ mice was significantly reduced compared with *Centlein*$^{+/+}$ and *Centlein*$^{+/-}$ mice ($n = 6$ independent experiments). Data are presented as mean ± SEM. A two-tailed Student's *t* test was performed, \*\*\**P* < 0.001, \*\*\*\**P* 0.0001. **c** The *Centlein*-null spermatozoa are headless. Single-sperm PNA (green) staining was performed using *Centlein*$^{+/+}$, *Centlein*$^{+/-}$, and *Centlein*$^{-/-}$ spermatozoa. Nuclei were stained with DAPI (blue). **d** Proportion of decapitated tails in *Centlein*$^{+/+}$, *Centlein*$^{+/-}$, and *Centlein*$^{-/-}$ caudal epididymis ($n = 6$ independent experiments). Data are presented as mean ± SEM. A two-tailed Student's *t* test was performed, \*\*\*\**P* < 0.0001. **e** Ultrastructure of *Centlein*$^{+/+}$ and *Centlein*$^{-/-}$ spermatozoa from caudal epididymis showing that the *Centlein*-null spermatozoa had no sperm head and contained a residual droplet of cytoplasm at the top of the flagellum with misarranged mitochondria inside. Nu: nuclear, M (red): mitochondrion, AX (blue): axoneme. The red asterisk indicates the missing microtubule doublets of axoneme in *Centlein*-null spermatozoa. Biologically independent mice were examined in three separate experiments with similar results. **f** Ultrastructure of the midpiece, principal piece, and end piece of *Centlein*$^{+/+}$ and *Centlein*$^{-/-}$ spermatozoa from caudal epididymis. M (red): mitochondrion, AX (blue): axoneme, Od (yellow): outer dense fibers. The red asterisk indicates the missing microtubule doublets of axoneme in *Centlein*-null spermatozoa. **g–i** Quantification ratio of abnormal axoneme in midpiece (**g**), principal piece (**h**), and the end piece (**i**) of *Centlein*$^{+/+}$ and *Centlein*$^{-/-}$ spermatozoa ($n = 3$ independent experiments). Data are presented as mean ± SEM. A two-tailed Student's *t* test was performed, \*\**P* < 0.01. Source data are provided as a Source Data file. Blue dots indicate *Centlein*$^{+/+}$ mice, green dots indicate *Centlein*$^{+/-}$ mice, and red dots indicate *Centlein*$^{-/-}$ mice.

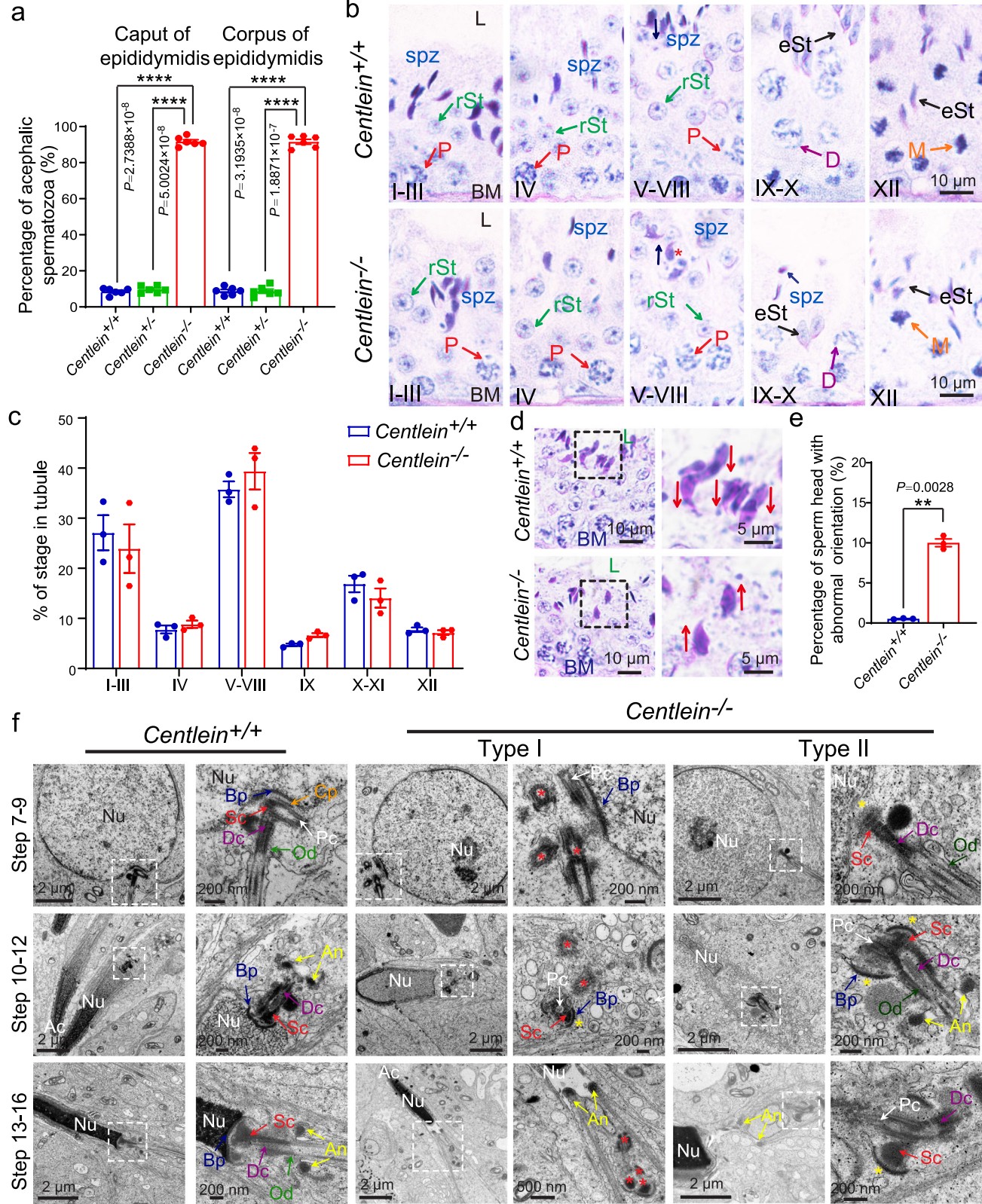

coupling apparatus of the spermatids, whereas it disappeared in spermatozoa (Fig. 5c and Supplementary Fig. 4).

In somatic cells, CENTLEIN localizes to the proximal ends of centrioles[37]. To visualize CENTLEIN in the spermatids, we co-stained the cells with the antibodies against CENTLEIN, CEP135, and the centriolar distal lumen protein CETN1/2 that has been reported to be localized on both Pc and Dc of the spermatids[35,36].

We found most of CENTLEIN and CEP135 signals were localized between two punctum signals of CETN1/2, indicating that CENTLEIN might localize on the Pc and Dc during spermatogenesis (Fig. 5d and Supplementary Fig. 5). Given that CENTLEIN complexed with SUN5 and PMFBP1, we performed immunofluorescent (IF) staining and found that CENTLEIN was indeed located in the HTCA region (Fig. 5e, f). In *Centlein*<sup>−/−</sup>

**Fig. 4 Centlein-deficient spermatids display impaired head-to-tail anchorage. a** The proportion of decapitated tails in Centlein$^{+/+}$, Centlein$^{+/-}$, and Centlein$^{-/-}$ corpus and caput epididymis (n = 6 independent experiments). Data are presented as mean ± SEM. A two-tailed Student's t test was performed, ****P < 0.0001. **b** PAS and hematoxylin staining were performed in Centlein$^{+/+}$ and Centlein$^{-/-}$ mouse. The mature sperm head could still be detected at stages IX–X in Centlein-deficient testes. The arrows indicate the orientation of the sperm heads. L: lumen, BM: basement membrane, P (red): pachytene spermatocyte, D (purple): diplonema spermatocyte, rST (green): round spermatid, eST (black): elongating spermatid, M (orange): meiotic spermatocyte, spz (blue): spermatozoa. Biologically independent mice were examined in three separate experiments with similar results. **c** The proportion of different stages of tubule cross-sections in Centlein$^{+/+}$ and Centlein$^{-/-}$ mice (n = 3 independent experiments). **d** Centlein-null spermatids have lost their orientation toward the basement membrane during spermiation in stage V–VIII seminiferous epithelia. The arrows indicate the orientation of the sperm heads. L: lumen, BM: basement membrane. **e** Quantification ratio of spermatids with abnormal orientation in Centlein$^{+/+}$ and Centlein$^{-/-}$ testes (n = 3 independent experiments). Data are presented as mean ± SEM. A two-tailed Student's t test was performed, **P < 0.01. **f** The coupling apparatus could not be tightly attached to the sperm head in Centlein-null mice. TEM analyses of the stepwise development of the coupling apparatus were performed in Centlein$^{+/+}$ and Centlein$^{-/-}$ testes. The red asterisk indicates destroyed coupling apparatus. The yellow asterisk indicates the gap between the nuclear envelope and coupling apparatus. Nu: nuclear, Ac: acrosome, Bp (blue): basal plate, Cp (orange): capitulum, Sc (red): segmented column, Pc (white): proximal centriole, Dc (purple): distal centriole, An (yellow): annulus, Od (green): outer dense fibers. Biologically independent mice were examined in three separate experiments with similar results. Source data are provided as a Source data file. Blue dots indicate Centlein$^{+/+}$ mice, green dots indicate Centlein$^{+/-}$ mice, and red dots indicate Centlein$^{-/-}$ mice.

spermatids, the HTCA localization of CENTLEIN was absent (Supplementary Fig. 6), and CEP135, CETN1/2, and PMFBP1 detached from the sperm heads (see below). Therefore, these results reveal that CENTLEIN is localized at the sperm HTCA during spermiogenesis.

**CENTLEIN intermediates SUN5 and PMFBP1 in HTCA.** We then attempted to determine the topology of SUN5 in the spermatid NE. First, we performed IF staining in spermatids using digitonin or Triton X-100 permeabilization, by which the digitonin selectively disrupts the plasma membrane leaving the NE membranes intact[39] and Triton X-100 permeabilizes all membranes. Antibody to the inner nuclear membrane LAP2[40] stained the NE only in Triton X-100-permeabilised spermatids (see below), whereas the antibody to the outer nuclear membrane marker SYNE1 labeled the spermatid NE after selective digitonin permeabilization (see below). Of importance, we noticed that the SUN5-labeling pattern in digitonin-permeabilized spermatids using the verified mouse anti-SUN5 antibody against the SUN5 SUN domain (Supplementary Fig. 7a, b) was indistinguishable from that in Triton X-100-treated spermatids (Supplementary Fig. 7c, d), in that both exhibited an NE staining, indicating that the SUN5 SUN domain being detected by the antibody is exposed to the spermatid cytosol. To further confirm it, an assay of in situ proteinase K digestions determining whether a protein resides within the lumen of a membrane bound organelle[41] was conducted in the round spermatids. Triton X-100 permeabilization in conjunction with proteinase K digestion resulted in degradation of cytoplasmic Tubulin, nuclear LAMIN B1, and SUN5 (Supplementary Fig. 7e), demonstrating that the in situ proteinase K digestions was reconstituted in round spermatids. When round spermatids permeabilized with digitonin followed proteinase K digestion, immunoblotting analysis revealed that the level of SUN5 probed with the mouse anti-SUN5 antibody against the SUN5 SUN domain was diminished (Supplementary Fig. 7e), which showed a similar phenomenon with the cytoplasmic Tubulin and nuclear LAMIN B1. Thus, by performing in situ proteinase K digestion assays and digitonin experiments, we demonstrated that the SUN5 SUN domain was exposed to cytoplasm of the spermatids, which, in turn, rendered the SUN5–CENTLEIN interaction in the spermatids.

To precisely map the interaction regions between CENTLEIN and its binding partners SUN5 and PMFBP1, reciprocal co-immunoprecipitation assays were carried out. By deletion analysis, we found that the 971–1396aa region of CENTLEIN was sufficient to bind SUN5 (Fig. 6a) and that the 601–1396aa region of CENTLEIN was necessary for its binding to PMFBP1

(Fig. 6b). Next, we mapped the CENTLEIN-binding site in SUN5 and PMFBP1. We found that the SUN domain of SUN5 (193–373aa) was sufficient to bind CENTLEIN (Fig. 6c), and two regions of PMFBP1, 1–282aa and 750–1023aa, were required for its binding to CENTLEIN (Fig. 6d). To further examine whether CENTLEIN directly binds to SUN5 and PMFBP1, we performed the GST pull-down experiments. We found that the SUN domain of SUN5 could directly interact with the 971–1396aa region of CENTLEIN (Fig. 6e), and the 750–1023aa region of PMFBP1 could directly bind to the 601–970aa region of CENTLEIN (Fig. 6f). All the aforementioned results point to CENTLEIN mediating an interaction between SUN5 and PMFBP1. To test this, we transfected HEK293T cells with GFP-tagged PMFBP1, FLAG-tagged SUN5, and MYC-tagged CENTLEIN and then performed anti-FLAG immunoprecipitations followed by western blotting (Fig. 6g). As shown in Fig. 6g, GFP-PMFBP1 was present in FLAG-SUN5 immunoprecipitate(s) only when MYC-tagged CENTLEIN was in the transfectant, indicating that CENTLEIN acts as a molecular linker between SUN5 and PMFBP1.

To examine whether CENTLEIN functions as a molecular linker in vivo, we performed the IF staining of SUN5 and PMFBP1 in Centlein$^{-/-}$ and Centlein$^{+/+}$ spermatids. We found that SUN5 was still visible in the connecting piece of the Centlein$^{-/-}$ spermatids (Fig. 7d, e), whereas PMFBP1 could not attach to the HTCA (Fig. 7d, f) attesting to the CEP135 and CETN1/2 staining patterns (Fig. 7a–c). Thus, CENTLEIN is essential for the connection between SUN5 and PMFBP1 in the HTCA. As the mouse models of acephalic spermatozoa are useful tools to dissect the defective formation and abortive maintenance of the HTAC, we further detected the localization of CENTLEIN in Sun5$^{-/-}$ and Pmfbp1$^{-/-}$ mice. We found that CENTLEIN could still attach to the implantation fossa of the sperm nucleus in Pmfbp1-null spermatids (Fig. 7g, h), whereas it was located in the cytoplasm in Sun5-null spermatids (Fig. 7i, j). Apparently, the three proteins are present at the HTAC in a hierarchical manner, i.e., SUN5–CENTLEIN–PMFBP1 (Fig. 8). Taken together, these data reinforced the inter-relationship among SUN5, CENTLEIN, and PMFBP1, accounting for their roles in anchoring sperm head to the tail.

## Discussion
SUN5, PMFBP1, and SPATA6 are three well-known HTCA proteins and localized in the neck of mammalian spermatids and spermatozoa, and the disruption of any of them leads to the production of acephalic spermatozoa[16,18,19,24]. Although they

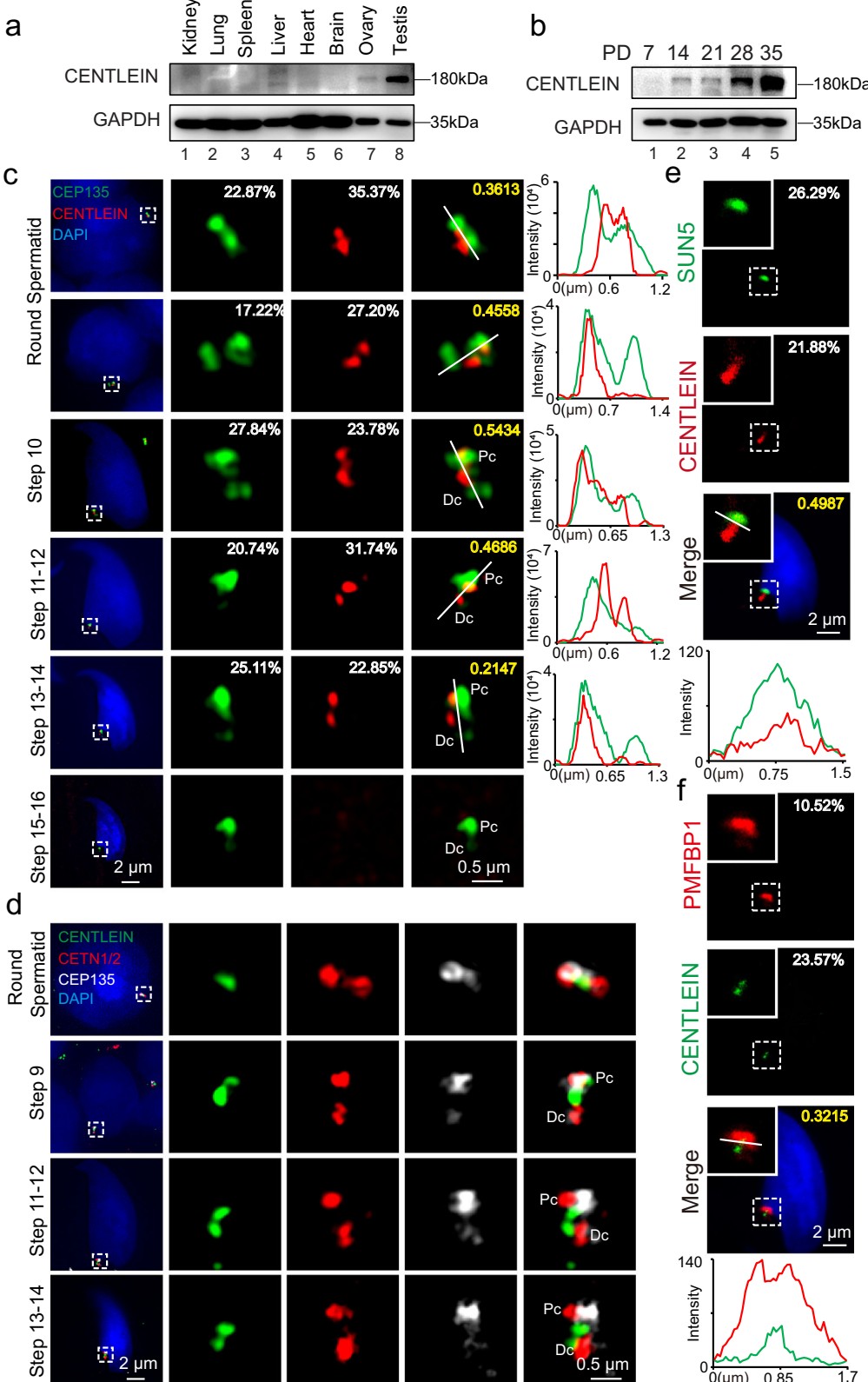

could form a sandwich-like structure on HTCA[19,24], they could not directly bind to each other, suggesting that some other structural proteins are still needed to integrate them into the HTCA. Here we found that CENTLEIN could directly interact with both SUN5 and PMFBP1, but not with SPATA6 (Figs. 1 and 6), and the mouse models also reveal that SUN5, CENTLEIN, and PMFBP1 are present at the HTAC in a hierarchical manner

(Figs. 5e, f and 7). Therefore, CENTLEIN works as a linker between SUN5 and PMFBP1 to maintain the integrity of HTCA (Fig. 8).

SUN5 belongs to the SUN domain-containing family of nuclear membrane proteins, which physically couple the nuclear lamina and cytoskeleton through the assembly of linker of nucleoskeleton and cytoskeleton complexes[42,43]. There were few works addressing

**Fig. 5 CENTLEIN is localized at the sperm head–tail coupling apparatus. a** CENTLEIN was predominately expressed in the testis. Immunoblotting of CENTLEIN was performed in the kidney, lung, spleen, liver, heart, brain, ovary, and testis. GAPDH served as the loading control. **b** The onset of CENTLEIN expression in P14 testes. Immunoblotting of CENTLEIN was performed in the testis at different postnatal days. GAPDH served as the loading control. **c** The localization of CENTLEIN at different developmental stages. Super-resolution microscopic images of CENTLEIN (red) and CEP135 (green) in testicular germ cells. Nuclei were stained with DAPI (blue). The pixel overlaps of CENTLEIN and CEP135 were quantified using the IMARIS software. Pearson's correlation coefficients were determined for the correlation of voxel intensity between CENTLEIN (red) and CEP135 (green) channels and are displayed in yellow. The Pearson's coefficients are 0.3613, 0.4558, 0.5434, 0.4686, and 0.2147. Line-scan analysis (white lines) using the ZEN software (Right). Pc proximal centriole, Dc distal centriole. **d** Super-resolution microscopic images of CENTLEIN (green), CETN1/2 (red), and CEP135 (white) in testicular germ cells. Nuclei were stained with DAPI (blue). Pc proximal centriole, Dc distal centriole. **e** The immunofluorescence analysis CENTLEIN (red) and SUN5 (green) was performed in testicular germ cells. Nuclei were stained with DAPI (blue). The pixel overlaps of CENTLEIN and SUN5 were quantified using the IMARIS software. Pearson's correlation coefficient was determined for the correlation of voxel intensity between CENTLEIN (red) and SUN5 (green) channels and are displayed in yellow. The Pearson's coefficient is 0.4987. Line-scan analysis (white line) using the LAS X software (Lower). **f** The immunofluorescence analysis for CENTLEIN (green) and PMFBP1 (red) was performed in testicular germ cells. Nuclei were stained with DAPI (blue). The pixel overlaps of CENTLEIN and PFMBP1 were quantified using the IMARIS software. Pearson's correlation coefficients were determined for the correlation of voxel intensity between CENTLEIN (green) and PMFBP1 (red) channels and are displayed in yellow. The Pearson's coefficient is 0.3215. Line-scan analysis (white line) using the LAS X software (Lower). The experiment was repeated three times independently with similar results (**a–f**).

the precise localization of SUN5 in sperm NE, due to the complicated localization of SUN5 in different studies[42,44]. Previously, we found the SUN5C (coiled-coil domain and the SUN domain of SUN5) region of SUN5 does not interact with KASH5 LR[21], indicating that SUN5 might be distinct from the classical SUN domain proteins. By using liquid chromatography–mass spectrometry and GST pull-down analysis, we found that SUN5 could directly bind to the DNAJB13 to facilitate SUN5 protein folding, which is required for the integrity of the HTCA[21]. In the present study, the digitonin and Triton X-100 permeabilization experiments showed that the SUN domain of SUN5 was exposed to the cytoplasm of the spermatids (Supplementary Fig. 7), and the SUN5 SUN domain could directly interact with the 971–1396aa region of CENTLEIN (Fig. 6e). Furthermore, the IF staining showed that CENTLEIN partially co-localized with SUN5 in the HTCA (Fig. 5e), and the absence of SUN5 perturbed the HTCA localization of CENTLEIN (Fig. 7 i, j). Thus, SUN5 might directly recruit CENTLEIN to the HTCA through its SUN domain.

One of the unique features of the spermiogenesis is that the Pc moves into the implantation fossa (also recognized as a normal lodging site for the Pc)[30,45], and some proteins may facilitate its attachment to the caudal end of the nucleus. SUN5 directly binds to CENTLEIN, a somatic centrosome protein localized at the proximal ends of the centrioles, but not the centriole distal lumen protein CETN2 (Fig. 1), indicating requisite association between the NE and Pc during spermatid differentiation. In the late step of spermiogenesis, CENTLEIN is no longer detectable, but the SPATA6 translocated from the manchette to the coupling apparatus[16], together with SUN5 and PMFBP1, to form a sandwich-like structure essential for the integrity of the HTCA[19].

Except acting as a molecular linker, CENTLEIN may also initiate the assembly of the ultrastructural components of the HTAC, as the ablation of CENTLEIN resulted in complete lack of the Scs and Cp in the early steps of round spermatids (Fig. 4f and Supplementary Fig. 3). The formation of HTCA is an intricate and complex process. In early round spermatids, the pair of centrioles localize to the caudal nuclear pole and expand the electron-dense material, part of which shows striation[32]. As the spermatids develop, the dense material surrounding the centrioles gradually becomes a well-organized structure, which can be clearly recognized as Scs, Cp, and Bp[24]. Although substructures of the HTCA were anatomically well defined and some centrosome proteins have been reported to be associated with the connection of the sperm head and tail[34], the molecular composition and assembly properties of the HTCA are poorly defined. Especially proteins that ensure the tight coupling to the nucleus await identification. Here we show that the centrosome protein,

CENTLEIN, is essential for the integrity of the HTCA (Fig. 4f and Supplementary Fig. 3). We have previously reported that in somatic cells CENTLEIN is localized to the proximal ends of centrioles and required for centrosome cohesion by mediating an interaction between C-Nap1 and Cep68[37]. During spermatogenesis, CENTLEIN is also partially co-localized with centrosomal protein CEP135 (Fig. 5c) and predominantly present between the two punctum signals of the centriole distal lumen proteins CETN1/2 (Fig. 5d). Apparently, CENTLEIN either acts as a cornerstone recruiting other centrosomal proteins as the "building blocks" for initiation of HTCA assembly or paralleled with other centrosomal proteins that are critical for HTCA formation. More efforts are definitely needed to shed light on the molecular basis of HTCA assembly during spermiogenesis.

HTCA formation anomalies might lead to acephalic spermatozoa syndrome, which results in headless spermatozoa and complete male infertility[18–20,24]. CENTLEIN binds to SUN5 and PMFBP1 (Figs. 1 and 6), both of which are encoded by two disease-causing genes of acephalic spermatozoa syndrome[19,20]. In both humans and mice, the disruption of either one results in the separation of sperm heads and tails during spermiogenesis and, in turn, leads to the decapitation of the spermatozoa and male infertility[18–20,24]. *CENTLEIN* should thus be a strong candidate for screening causative genes for human acephalic sperm conditions.

## Methods
**Animals**. The *Sun5*[−/−] and *Pmfbp1*[−/−] mice have been reported previously[18,19]. The *Centlein*[−/−] mice were generated by applying the CRISPR-Cas9 system. The T7 promoter and the guiding sequence were added to the single-guide RNA (sgRNA) by PCR amplification using the following primers: *Centlein*-For: 5′-GTA GCT GTG GTG GCA TCT CTG GG-3′ and *Centlein*-Rev:5′-TTC TTT ATG AAG CGC TGC GTT GG-3′. B6D2F1 (C57BL/6J × DBA2/J) female mice and ICR female mice were used as embryo donors and foster mothers, respectively. Superovulated female B6D2F1 mice (6–8 weeks old) were mated with B6D2F1 stud males, and the fertilized embryos were collected from the oviducts. Cas9 mRNA (20 ng) and sgRNA (10 ng) were injected into the cytoplasm of fertilized eggs with well-recognized pronuclei in M2 medium (M7167, 50 ml, Sigma). The injected zygotes were cultured in KSOM (modified simplex-optimized medium, Millipore) with amino acids at 37 °C under 5% $CO_2$ in air and then transferred into the uterus of pseudopregnant ICR females. The genotyping primers were as follows (Supplementary Table 1): forward (*Centlein*-491F), and reverse (*Centlein*-1353R) for WT allele (863 bp); forward (*Centlein*-491F) and reverse (*Centlein*-217278R) for *Centlein* knockout allele (700 bp). Mice were housed in the same animal facility on a 12-h reverse light/dark cycle. The animal facility was maintained at a temperature of 22–24 °C with 40–60% humidity. Animal experiments were conducted under the protocol and approval (IOZ20170079) of the Animal Care and Use Committee of the Institute of Zoology, Chinese Academy of Sciences, China. The present study is compliant with all relevant ethical regulations regarding animal research.

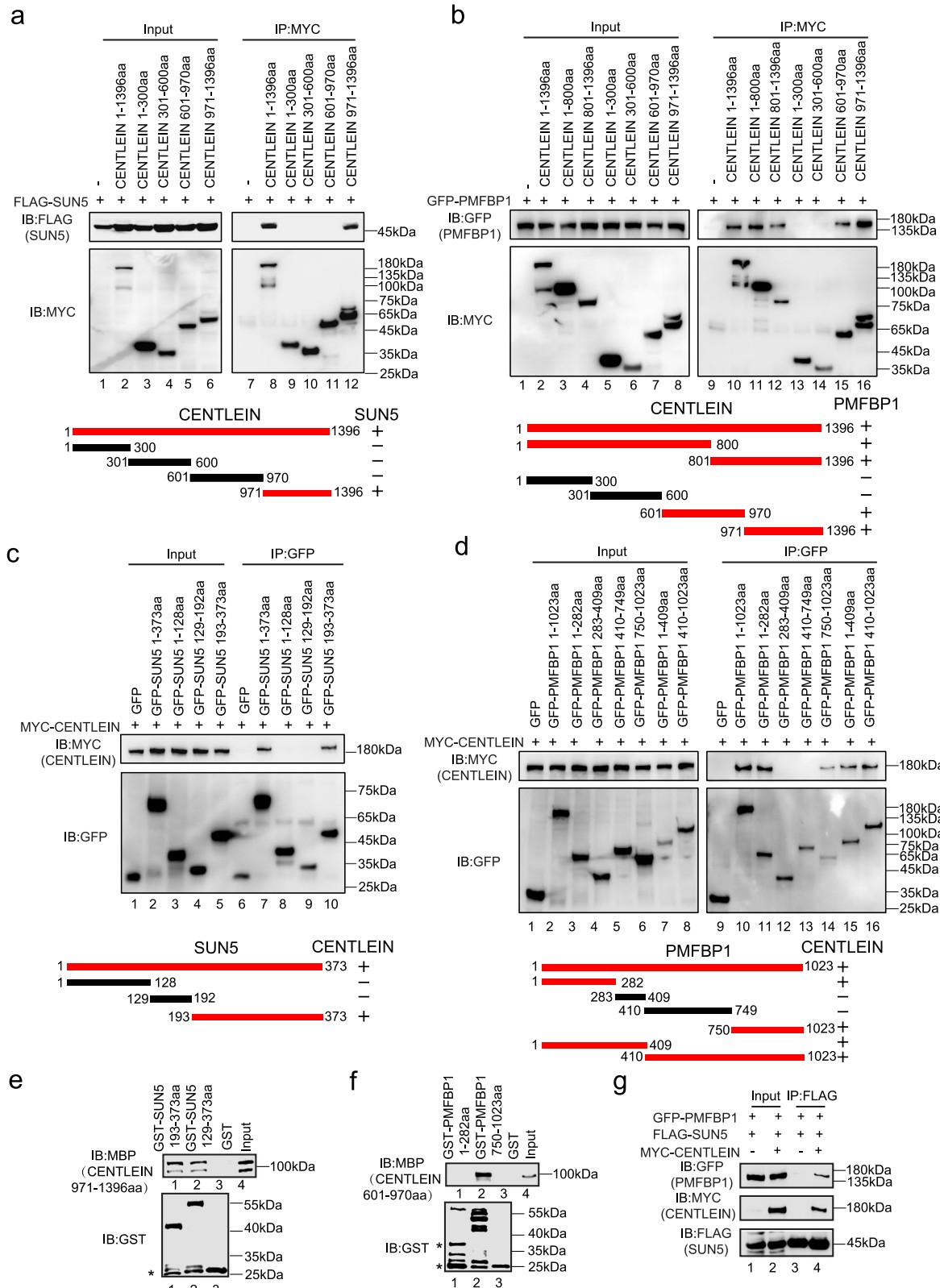

**Plasmids.** *Sun5* and *Spata6* were cloned into pRK-FLAG vector and P*mfbp*1 was cloned into pEGFP-C1 vector[19]. Full-length mouse *Centlein* was obtained from mouse testis cDNA and cloned into the vectors pEGFP-C1 and pCMV-MYC. *Centlein*, *Cep68*, *Nek2a*[37], *Bbs4*, *Cp110*, *Cpap*, *Cetn2*, and *Plk1* were obtained from HeLa cell cDNA and cloned into the pEGFP-C1 vector using the Phanta® Max Super-Fidelity DNA Polymerase (P505, Vazyme). The truncated mutants of *Centlein* were cloned into the vectors pCMV-MYC and pMAL-c2x using the Clon Express Ultra One Step Cloning Kit (C115, Vazyme). The truncated

mutants of *Pmfbp1* and *Sun5* were cloned into the vectors pEGFP-C1 and pGEX-4T-3.

**Antibodies.** Rat monoclonal anti-CENTLEIN antibody (aa1–280) was generated by Absea Biotechnology Ltd (Beijing, China). Rat clone 9F8 has been used at a 1:20 dilution for IF and 1:500 for western blotting. Rabbit anti-GFP antibody (50430-2-AP, Proteintech), rabbit anti-FLAG antibody (20543-1-AP, Proteintech), mouse

**Fig. 6 CENTLEIN directly interacts with SUN5 and PMFBP1. a** Amino acids 971–1396 of CENTLEIN are necessary to bind to SUN5. HEK293T cells were co-transfected with FLAG-SUN5 and the indicated fragments of MYC-CENTLEIN, immunoprecipitated with anti-MYC antibody, and then immunoblotted with FLAG and MYC antibodies, respectively. +, red, interaction; −, black, no interaction. **b** Amino acids 601–970 and 971–1396 of CENTLEIN are necessary to bind to PMFBP1. HEK293T cells were co-transfected with GFP-PMFBP1 and the indicated fragments of MYC-CENTLEIN, immunoprecipitated with anti-MYC antibody, and then immunoblotted with MYC and GFP antibodies, respectively. +, red, interaction; −, black, no interaction. **c** The SUN domain of SUN5 is necessary to bind to CENTLEIN. HEK293T cells were co-transfected with MYC-CENTLEIN and the indicated fragments of GFP-SUN5, immunoprecipitated with anti-GFP antibody, and then immunoblotted with GFP and MYC antibodies, respectively. +, red, interaction; −, black, no interaction. **d** Amino acids 1–282 and 750–1023 of PMFBP1 are necessary to bind to CENTLEIN. HEK293T cells were co-transfected with MYC-CENTLEIN and the indicated fragments of GFP-PMFBP1, immunoprecipitated with anti-GFP antibody, and then immunoblotted with MYC and GFP antibodies, respectively. +, red, interaction; −, black, no interaction. **e** The SUN domain of SUN5 directly bind the 971–1396aa region of CENTLEIN. GST-SUN5 129–373aa and GST-SUN5 193–373aa were purified and used to pull down MBP-CENTLEIN 971–1396aa; GST was used as a control. Asterisks in the bottom panel indicate GST products cleaved from the fused proteins. **f** Amino acids 750–1023 of PMFBP1 directly bind the 601–970aa region of CENTLEIN. GST-PMFBP1 1–282aa and GST-PMFBP1 750–1023aa were purified and used to pull down MBP-CENTLEIN 601–970aa; GST was used as a control. Asterisks in the bottom panel indicate GST products cleaved from the fused proteins. **g** CENTLEIN acts as a molecular linker between SUN5 and PMFBP1. GFP-PMFBP1, FLAG-SUN5, and empty vector or MYC-CENTLEIN plasmids were co-transfected into HEK293T cells. Twenty-four hours after transfection, cells were collected for immunoprecipitation (IP) with anti-FLAG antibody and analyzed with FLAG, MYC, and GFP antibodies, respectively. The experiment was repeated three times independently with similar results (**a–g**).

anti-FLAG antibody (Clone M2, F3165, Sigma-Aldrich), mouse anti-MYC antibody (M192-3, MBL International), mouse anti-MBP antibody (66003-1-Ig, Proteintech), mouse anti-GST antibody (M20007L, Abmart), and rabbit anti-α-Tubulin (AC007, ABclonal) were used at a 1:2000 dilution for western blotting. Mouse anti-Lamin B1 antibody (66095-1-Ig, Proteintech) was used at a 1:1000 dilution for western blotting. Mouse anti-GAPDH antibody (ab1019t, AmeriBiopharma) was used at a 1:5000 dilution for western blotting. Rabbit anti-PMFBP1 polyclonal antibody against aa673–1022 (1:100, 17061-1-AP, Proteintech), rabbit anti-SUN5 polyclonal antibody against aa1–379 (1:100, 17495-1-AP, RRID: AB_1939754, Proteintech), mouse anti-γ-tubulin (1:200, TU-30, sc-51715, Santa Cruz), rabbit anti-CEP135 antibody against C-terminal (1:1500, A02C0240, Blue Gene), rabbit anti-LAP2 antibody (1:200, 14651-1-AP, Proteintech), and rabbit anti-SYNE1 antibody (1:20, HPA019113, Atlas Antibodies) were used for IF. The mouse SUN5 antibody against the SUN5 SUN domain (aa193–373) for IF (1:100) and western blotting (1:100) was in-house-generated. GFP-Booster (1:200, gba488-100, Chromotek) for IF was a gift from Juntao Gao (Tsinghua University). A mouse anti-CETN1/2 antibody against C-terminus (1:100, clone 20H5, 04-1624, Millipore) was covalently coupled to Alexa Fluor 594, using an APEX Antibody Labeling Kit (A10474, Invitrogen) for IF. The secondary antibodies used were horseradish peroxidase (HRP)-conjugated goat anti-mouse IgG (1:4000, ZB-2305, Zhong Shan Jin Qiao), HRP-conjugated goat anti-rabbit IgG (1:4000, ZB-2301, Zhong Shan Jin Qiao), goat anti-rabbit FITC (1:200, ZF-0311, Zhong Shan Jin Qiao), goat anti-rabbit TRITC (1:200, ZF-0316, Zhong Shan Jin Qiao), goat anti-mouse TRITC (1:200, ZF-0313, Zhong Shan Jin Qiao), donkey anti-rabbit Cy5 (1:200, 711-175-152, Jackson ImmunoResearch), Alexa Fluor 594 goat anti-rat IgG (1:1500, A11007, Invitrogen), and Alexa Fluor 488 goat anti-rat IgG (1:1500, A11006, Invitrogen).

**Cell culture**. HEK293T cells obtained from ATCC (CRL-3216) were cultured in high-glucose Dulbecco's Modified Eagle Medium (DMEM, SH30243.01, Hyclone) supplemented with 10% fetal bovine serum (F2442, Sigma) and 1% penicillin–streptomycin (SV30010, Hyclone) at 37 °C in a humidified incubator containing 5% CO₂. The cells were tested negative for mycoplasma contamination.

**Immunoprecipitation**. Transfected HEK293T cells were lysed in ice-cold ELB buffer [50 mM HEPES, 250 mM NaCl, 0.1% NP-40, 1 mM phenylmethanesulfonylfluoride (PMSF; P7626, Sigma), and complete EDTA-free protease inhibitor cocktail (04693132001, Roche)]. The lysates were centrifuged at 12,000 × g for 10 min and the supernatant was precleared by incubation with 50 μl of protein-G–Sepharose (CW0012A, Cowin Biotech) at 4 °C for 3 h. After that, the supernatant was incubated with 2 μg antibody at 4 °C for an additional 3 h, followed by addition of 20 μl Dynabeads–protein-G (10004D, Invitrogen) and incubated at 4 °C overnight. Beads were washed four to six times using ELB buffer, followed by heating of the precipitated material in sodium dodecyl sulfate–polyacrylamide gel electrophoresis (SDS-PAGE) sample buffer and immunoblotting analysis.

**Immunoblotting**. Proteins obtained from lysates or immunoprecipitates were separated by SDS-PAGE and transferred to polyvinylidene difluoride membranes (IPVH00010, Millipore). The membranes were then incubated in TBS-T (10 mM Tris–HCl pH|7.4, 150 mM NaCl, and 0.1% Tween-20) containing 5% non-fat milk at room temperature for 1 h and stained with the appropriate primary and secondary antibodies. After final washes with TBS-T, the membranes were developed by using ECL prime western blotting detection reagent (RPN 2232, GE Healthcare Life Sciences). The images were taken immediately using Tanon 4100 imaging system with the GelCap 5.6 software.

**Assessment of fertility**. Fertility was tested in the male mice of the different genotypes (8 or 9 weeks old). Each male mouse was caged with two WT CD1 females (7–8 weeks), and vaginal plug was checked every morning. The plugged female was separated and single caged, and the pregnancy results were recorded. If a female did not generate any pups by day 22 postcoitus, it was deemed as not pregnant and euthanized to confirm that result. Each male underwent six cycles of the above breeding assay with different females.

**Epididymal sperm count**. The caudal epididymis was dissected from 8-week-old mice. Spermatozoa were squeezed out from the caudal epididymis and incubated for 30 min at 37 °C under 5% CO₂. The incubated sperm medium was then diluted 1:10. A cover slip was placed on the hemocytometer before a drop with 10 μl of diluted caudal epididymal sperm solution was loaded under the cover slip. The hemocytometer was placed under the Primo Star microscope (Zeiss) and viewed under ×400 magnification. The microscope was not equipped with camera or software, and the light source of the microscope is LED: 3 W 3200k. Sperm count was done by counting 4 × 4 squares (horizontally or vertically) using the hemocytometer and calculated using the formula: Sperm count = total no. of sperm in 5 squares × 50,000 × dilution multiple (cells/ml). Counting was only done for sperm tails that was found within the square areas. The total number of sperm was counted and the mean was calculated from three counts. Six independent experiments were performed. The data were then analyzed with GraphPad Prism 7.

**Tissue collection and histological analysis**. The caudal epididymis from at least three mice for each genotype were dissected immediately after euthanasia, fixed in 4% (mass/vol) paraformaldehyde (PFA; P1110, Solarbio) for up to 24 h, stored in 70% (vol/vol) ethanol, and embedded in paraffin. The 5 μm sections were prepared and mounted on glass slides. After deparaffinization, slides were stained with H&E for histological analysis. For PAS– hematoxylin staining, testes were fixed by perfusing mice with Bouin's fixatives (16045-1, Polysciences). After deparaffinization, slides were stained with PAS and hematoxylin. Stages of seminiferous epithelium cycle and spermatid development were determined.

**Isolation of the testicular germ cells**. The mouse testis was dissected and fixed with 2% PFA in 0.05% PBST (phosphate-buffered saline (PBS) with 0.05% Triton X-100) at room temperature for 10 min. The fixed sample was placed on a slide glass and squashed by placing a cover slip and pressing it down gently. The slides were immersed in liquid nitrogen and stored at −80 °C.

**IF and image analysis**. The spermatozoa were spread on glass slides for morphological observation or immunostaining. After air drying, spermatozoa were fixed in 4% PFA at room temperature for 10 min, and slides were washed with PBS three times and then treated with 0.5% Triton X-100 for 10 min, rinsed in PBS three times, and blocked in 5% bovine serum albumin (BSA, AP0027, Amresco) in PBS for 30 min. For permeabilizing with digitonin, fixed isolated germ cells were washed in ice-cold PBS and treated with 0.004% digitonin (CAS: 11024-24-1) in PBS for 10 min on ice. The primary antibody was added to the sections and incubated at 4 °C overnight, followed by incubation with secondary antibody at 1:200 for 1 h at 37 °C. The nuclei were stained with DAPI (D3571, Life Technologies). HEK293T cells were transfected using Lipofectamine 2000 reagent (11668019, Invitrogen) according to the manufacturer's protocol. Cells were analyzed 8 h post transfection. Cells were grown on cover slips, rinsed in PBS and fixed in −20 °C methanol for 8 min, washed with PBS three times and blocked in 2% bovine serum albumin in PBS for 30 min, then incubated with primary antibodies overnight at 4 °C, followed by incubation with secondary antibody at 1:200 for 30

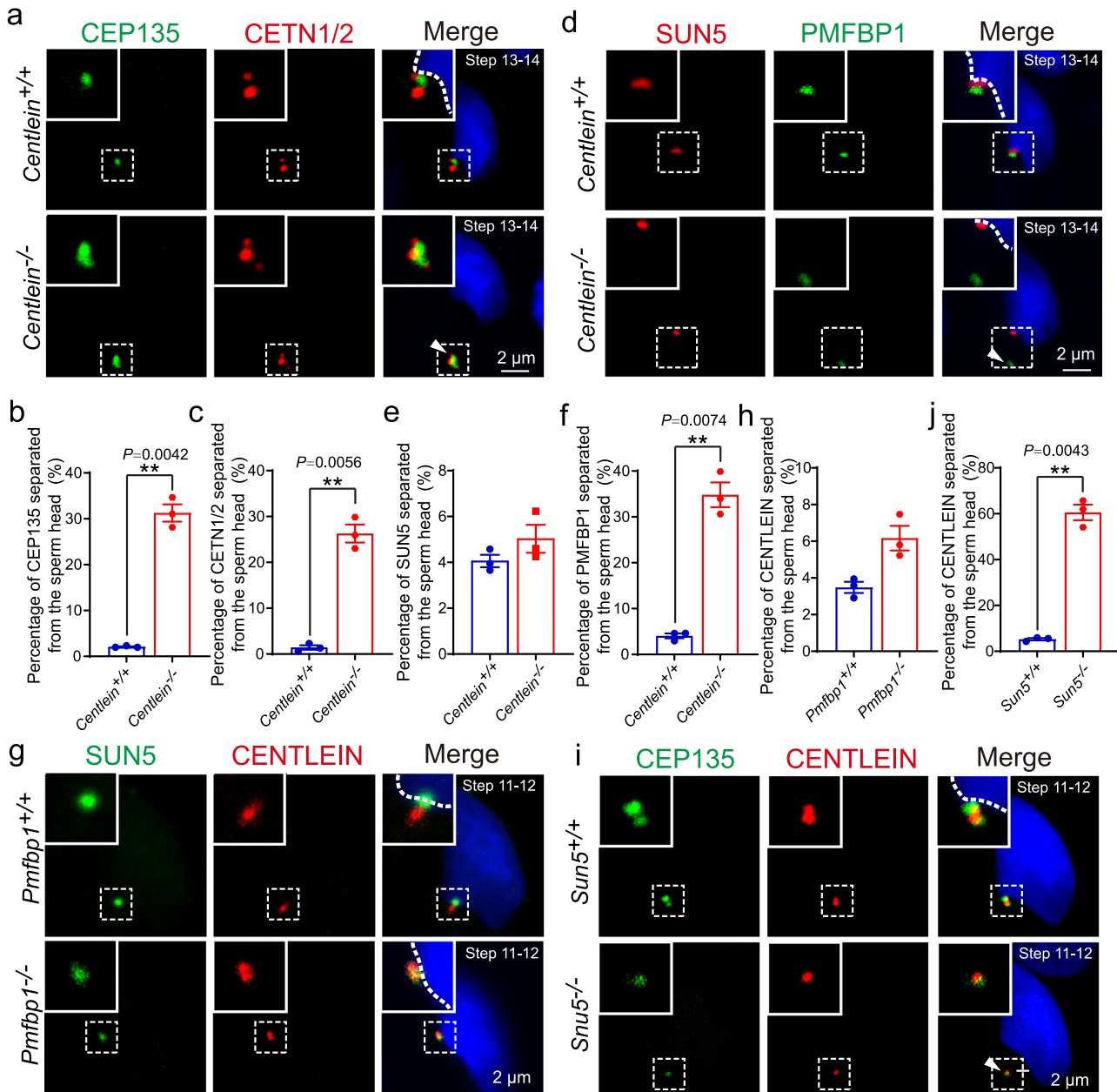

**Fig. 7 CENTLEIN cooperates with SUN5 and PMFBP1 to connect the sperm tail to its head. a** Immunofluorescence analysis for CEP135 (green) and CETN1/2 (red) was performed in *Centlein*$^{+/+}$ and *Centlein*$^{-/-}$ testicular germ cells. Nuclei were stained with DAPI (blue). The arrow head indicates detached CEP135 and CETN1/2 from sperm head. **b, c** Quantification ratio of CEP135 (**b**) and CETN1/2 (**c**) separated from the sperm head >1.5 μm in *Centlein*$^{+/+}$ and *Centlein*$^{-/-}$ mice (*n* = 3 independent experiments). Blue dots indicate *Centlein*$^{+/+}$ mice and red dots indicate *Centlein*$^{-/-}$ mice. **d** Ablation of *Centlein* impairs the localization of PMFBP1 to the coupling apparatus. Immunofluorescence analysis for PMFBP1 (green) and SUN5 (red) was performed in *Centlein*$^{+/+}$ and *Centlein*$^{-/-}$ testicular germ cells. Nuclei were stained with DAPI (blue). The arrow head indicates detached PMFBP1 from sperm head. **e, f** Quantification ratio of SUN5 (**e**) and PMFBP1 (**f**) separated from the sperm head >1.5 μm in *Centlein*$^{+/+}$ and *Centlein*$^{-/-}$ mice (*n* = 3 independent experiments). Blue dots indicate *Centlein*$^{+/+}$ mice and red dots indicate *Centlein*$^{-/-}$ mice. **g** Disruption of *Pmfbp1* has no influence on the localization of CENTLEIN to the coupling apparatus. Immunofluorescence analysis for CENTLEIN (red) and SUN5 (green) was performed in *Pmfbp1*$^{+/+}$ and *Pmfbp1*$^{-/-}$ testicular germ cells. Nuclei were stained with DAPI (blue). **h** Quantification ratio of CENTLEIN separated from the sperm head >1.5 μm in *Pmfbp1*$^{+/+}$ and *Pmfbp1*$^{-/-}$ mice (*n* = 3 independent experiments). Blue dots indicate *Pmfbp1*$^{+/+}$ mice and red dots indicate *Pmfbp1*$^{-/-}$ mice. **i** Lack of *Sun5* perturbs the localization of CENTLEIN to the coupling apparatus. Immunofluorescence analysis for CENTLEIN (red) and CEP135 (green) was performed in *Sun5*$^{+/+}$ and *Sun5*$^{-/-}$ testicular germ cells. Nuclei were stained with DAPI (blue). The arrow head indicates detached CENTLEIN and CEP135 from the sperm head. **j** Quantification ratio of CENTLEIN separated from the sperm head >1.5 μm in *Sun5*$^{+/+}$ *and Sun5*$^{-/-}$ mice (*n* = 3 independent experiments). Blue dots indicate *Sun5*$^{+/+}$ mice and red dots indicate *Sun5*$^{-/-}$ mice. Data in **b, c, e, f, h, j** are presented as mean ± SEM. A two-tailed Student's *t* test was performed, \*\**P* < 0.01.

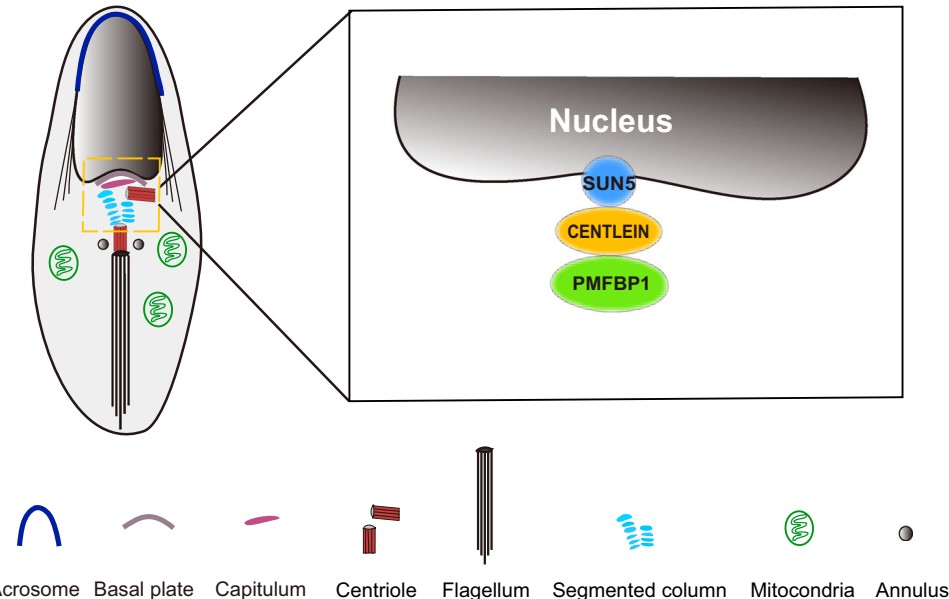

**Fig. 8 A proposed model for the role of CENTLEIN in integration of sperm head to the tail.** CENTLEIN works as a linker between SUN5 and PMFBP1 to maintain the integrity of HTCA.

min at room, then mounted on slides with ProLong Gold (P36931, Invitrogen) containing DAPI to stain DNA. The IF images were taken immediately using an SP8 microscope (Leica) equipped with a ×63 oil immersion objective or ELYPA S.1 microscope (Zeiss) equipped with a ×100 oil immersion objective. The imaging software of SP8 and ELYPA S.1 microscopes is the Leica Application Suite X (LAS X) 3.0 and ZEN 2.3 softwares, respectively. Structured illumination microscopy (SIM) data were collected using an ELYPA S.1 microscope (Zeiss), an sCMOs camera, and 405, 488, 561, and 640 nm excitation lasers. Hundred nanometer-thick Z sections were acquired in three-dimensional (3D) SIM mode generating 7 images per plane (5 phases, 3 angles) as a raw image, which was reconstructed to generate a super-resolution image. Channel alignment was conducted using calibrated file generated from 200 nm diameter tetra-spectral fluorescent spheres (Life Technologies). Images were exported to be further analyzed in the IMARIS 9.0.2, ZEN 2.3 lite, or LAS X 3.0 softwares. Co-localization analysis was performed utilizing the IMARIS software. The level of co-localization in the 3D volume was measured as a percentage of volume of the channel co-localized. Using IMARIS surface module and background subtraction function, we calculated the volume of green and red channels, respectively. Using IMARIS Coloc module, we set a threshold and calculated the volume of a new extracted co-localization channel. The level of co-localization was expressed as the percentage of the co-localization volume to the volume of the green or red channel, respectively. A second measure of the intensity of co-localization between two signals was obtained by calculating the correlation between the intensities of the co-localized 3D pixels (Pearson's correlation coefficient). The extent of co-localization of two labels was measured using the Coloc module. The intensity threshold in both channels was automatically determined. The Pearson's correlation coefficient lies between +1 and −1, with positive values indicating a direct correlation and values near 0 indicating no correlation. Relative pixel intensities of fluorescence were analyzed by line-scan analysis using the ZEN software or LAS X software. The position of line-scan is indicated by a white line on the merged image. Using Profile module and line-scan function, we measured intensity of two fluorescence signals (red and green) across the white line.

**Transmission electron microscopy.** The adult mouse testes and caudal epididymis were dissected and fixed with 2.5% (vol/vol) glutaraldehyde in 0.1 M cacodylate buffer overnight. After washing in 0.1 M cacodylate buffer, tissues were cut into small pieces of approximately 1 mm$^3$ and immersed in 1% OsO$_4$ for 1 h at 4 °C. Then the samples were dehydrated through a graded acetone series and embedded in resin. Ultrathin sections were cut on an ultramicrotome, stained with uranyl acetate and lead citrate, and observed using a JEM-1400 transmission electron microscope (JEOL) operating at 80 kV. Images were acquired at 11 million pixels using CCD camera (Gatan 832).

**Purification of mouse spermatogenic cells.** Spermatogenic cells were purified using a method previously described[46]. Briefly, testes from adult mice were removed and decapsulated. The seminiferous tubules were tore into small pieces and incubated in 10 ml DMEM (12100-046, Gibco) containing 1 mg/ml collagenase (C5138, Sigma) and 1 mg/ml hyaluronidase (H3506, Sigma) at 37 °C for 5 min with

gentle shaking. After pipetting, the dispersed seminiferous tubules and cells were incubated at 37 °C for 5 min with gentle shaking. Then the cells were collected by centrifugation at $200 \times g$ for 5 min at 4 °C, washed once with PBS, resuspended in 10 ml PBS containing 0.25% Trypsin and 1 mg/ml DNase I, and incubated at 37 °C for 5 min with gentle shaking. Cells were collected by centrifugation at $600 \times g$ for 5 min and washed with 10 ml DMEM containing 0.5% BSA. After filtration through a 40 μm Nylon Cell Strainer, the cells were separated by 3 h of velocity sedimentation at unit gravity, using a 2–4% BSA gradient in DMEM. The cell fractions were bottom-loaded in a volume of 300 ml. The cell type and purity in each fraction were assessed using light microscope based on their diameters and morphological characteristics. Only fractions with the expected cell type and purity (≥90%) were pooled together.

**In situ proteinase K digestion.** Spermatogenic cells were purified by the above method and subjected to in situ proteinase K digestion[41]. In brief, after two rinses with ice-cold PBS, one lot was incubated in 4 μg/ml proteinase K (EO0491, Thermo Scientific) in KHM buffer (110 mM KOAc, 20 mM Hepes, pH 7.4 and 2 mM MgCl$_2$) for 45 min at room temperature. The second lot was permeabilized with 24 μM of ice-cold digitonin in KHM for 15 min followed by 4 μg/ml proteinase K digestion in KHM for 45 min at room temperature. The third lot was incubated with 4 μg/ml proteinase K in KHM containing 0.5% Triton X-100 for 45 min. Subsequently, PMSF was added to all lots to a final concentration of 40 μg/ml. Cells were then washed in KHM and lysed in 0.4% SDS, 2% Triton X-100, 400 mM NaCl, 50 mM Tris–HCl, pH 7.4, 40 μg/ml PMSF, 1 mM dithiothreitol, and protease inhibitors (04693132001, Roche) by passing through a 22-gauge needle before centrifugation for 10 min at $16,000 \times g$. Proteins were then probed with specific antibodies by western blotting.

**GST pull-down experiments.** GST-tagged and MBP-tagged fragments were expressed in *Escherichia coli*. GST-tagged proteins were incubated with Glutathione Sepharose 4B (17-0757-01, GE Healthcare) at 4 °C for 2 h. The GST protein was used as a negative control. The GST beads were centrifuged and washed with a high-salt buffer (50 mM Tris [pH 7.4], 150 mM potassium chloride, 2 mM magnesium chloride, 0.1% Triton X-100). GST-tagged proteins were added, and the beads were incubated at 4 °C for 2 h. The beads were centrifuged and washed three times with the high-salt buffer, and SDS loading buffer was added. The samples were analyzed by immunoblotting with anti-GST and anti-MBP antibodies.

**Statistics and reproducibility.** All data are presented as mean ± SEM. The statistical significance of the differences between the mean values for the different genotypes was measured by Student's *t* test with a paired, two-tailed distribution using the analysis softwares GraphPad Prism 7 and Microsoft Excel 2010. The data were considered significant when $*P < 0.05$, $**P < 0.01$, $***P < 0.001$, and $****P < 0.0001$. All results shown in the study are representative of at least three independent experiments with similar results.

**Reporting summary**. Further information on research design is available in the Nature Research Reporting Summary linked to this article.

## Data availability
The authors declare that all data supporting the findings of this study are available within the article and its Supplementary Information files or from the corresponding author on reasonable request. Source data are provided with this paper.

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

## Acknowledgements
We thank Juntao Gao (Tsinghua University, Beijing, China) for kindly providing the GFP-Booster antibody. We thank Pengyan Xia (State Key Laboratory of Membrane Biology, Institute of Zoology, Chinese Academy of Sciences, China) for his help in making TEM sample. We thank Dr. Fei Gao (Institute of Zoology, Chinese Academy of Sciences, China) for critical reading of the manuscript. This work was supported by the National Key R&D Program of China (Grant No. 2018YFC1004202), the National Natural Science Foundation of China (31872839 to L.Y., 31771501 to C.L.), the National Science Fund for Distinguished Young Scholars (81925015 to W.L.), and the Youth Innovation Promotion Association of CAS (2018109).

## Author contributions
Y.Z. and C.L. performed most of the experiments. B.W. and L.L. performed some immunofluorescence experiments. W.L. and L.Y. supervised the project and wrote the manuscript.

## Competing interests

The authors declare no competing interests.
