## [Peer Review File · Nature Communications]

Reviewers' Comments:

Reviewer #1:

Remarks to the Author:

This work investigates how SUN5 and PMFBP1 function together in the head-to-tail coupling apparatus (HTCA) during spermatogenesis. This information is important, as mutations in the genes that encode SUN5 and PMFBP1 were previously associated with acephalic spermatozoa syndrome. The authors provide evidence to support the hypothesis that SUN5 and PMFBP1 indirectly interact with each other through the centrosomal protein CENTLEIN. They show that male mice lacking CENTLEIN are sterile due to their production of acephalic sperm. Transmission electron microscope-based analysis of the ultrastructure of sperm produced by CENTLEIN-null mice revealed defects in the assembly of the segmented column and the capitulum of the HTCA at early stages of spermatogenesis as well as the detachment of the anomalous segmented column and basal plate from spermatid nuclear envelopes. Based on confocal images of immunofluorescently stained sperm isolated from CENTLEIN-, PMFBP1-, or SUN5-null mice, the authors conclude that CENTLEIN/SUN5/PMFBP1 function during HTCA assembly and the resulting integration of the sperm head and tail. These results suggest that mutations in CENTLEIN may also be associated with acephalic spermatozoa syndrome in humans. Overall, I feel that this manuscript is well written as well as tackles an interesting and important question in the field of spermatogenesis. However, I am not convinced that the data presented in this manuscript fully supports the authors' conclusions. Moreover, I do not feel that this work represents enough of a mechanistic advance for the field. Thus, I cannot recommend that this manuscript be accepted for publication in its current state. Below, I will briefly outline the major and minor issues that I feel that the authors must address in order for their manuscript to be accepted for publication.

Major issues.

- 1) The authors need to better describe how they envision that SUN5 functions in the HTCA. SUN5 is a member of the SUN family of inner nuclear membrane proteins, yet the authors fail to mention this important fact anywhere in their manuscript. Nor do they mention that SUN proteins are core components of the linker of nucleoskeleton and cytoskeleton complex due to their ability to form a transluminal interaction with the KASH/nesprin family of outer nuclear membrane proteins. Based on this information, do the authors propose that the normally nucleoplasmic N-terminus of SUN5 resides on the outer nuclear membrane where it is able to interact with CENTLEIN? Alternatively, does SUN5 function to recruit a KASH/nesprin protein to the outer nuclear membrane of the sperm nucleus that is capable of interacting with CENTLEIN? The authors need to discuss these opposing models in their manuscript.
- 2) The authors need to provide a brief description of what is known about the cellular functions of the SPATA6 and PMFBP1 proteins. Without this background information, it is difficult to understand how these proteins might be working together with SUN5 and CENTLEIN in the HTCA.
- 3) Continuing along the theme of mechanism of SUN5 function described above, the authors really need to map the CENTLEIN binding site in SUN5. They also should map the SUN5 and PMFBP1 binding sites on CENTLEIN. Moreover, they need to test if the SUN5-CENTLEIN interaction were direct or not using recombinant proteins. Despite what the authors state in line 95 of the Introduction, the ability of two proteins to be co-immunoprecipitated does not prove that they are able to directly interact with each other. While I understand that the pandemic makes it difficult to perform additional experiments at this time, I strongly feel that a deeper mechanistic understanding of the CENTLEIN/SUN5/PMFBP1 interaction is needed for this manuscript to be suitable for publication at Nat Commun.
- 4) The authors fail to mention the source of the cDNA constructs that encode the EGFP-tagged centrosome proteins used in Figure 1. Are these previously described constructs? Are they functional? If they are not, the authors should provide representative images of the subcellular localization of these EGFP-tagged centrosome proteins in the HEK293T cells to demonstrate that these constructs are properly localized to the centrosome.
- 5) The authors should provide a better description of the different stages of spermatogenesis for their readers who may not be experts. Without this information, the manuscript is difficult to follow.
- 6) Page 11, lines 245-247: The authors state "PMFBP1 could not attach to the implantation fossa of the sperm nucleus...Thus CENTLEIN is essential for the connection between SUN5 and PMFBP1 in the HTCA". However, they do not provide enough experimental evidence to support these claims.

The authors do not know if PMFBP1 cannot interact with the implantation fossa of the sperm nucleus nor do they know that CENTLEIN is essential for the SUN5-PMFBP1 interaction.

7) Figure 1.

a. It appears that FLAG-SUN5 can also interact with the following GFP-tagged centrosome proteins: BBS4m, NEK2, CPAP, and P50. While these interactions are clearly less pronounced than what was observed with CENTLEIN, they are completely ignored in the manuscript. Are these 4 proteins known to be involved in spermatogenesis? This needs to be discussed.

b. The authors need to indicate the position of their molecular weight markers on their Western blots.

c. The authors should use asterisks to label the bands that they want their readers to focus on in the GFP blot shown in Figure 1A.

d. What are the big black bands found in the FLAG blot shown in Figure 1G? Are they IgG light or heavy chains? The amount of over-saturation of these big black bands makes it difficult to see the immunoprecipitated FLAG-SPATA6 band.

8) Figure 2.

a. The authors need to indicate the 5' and 3' ends of the DNA sequences shown in Figure 2A.

b. The authors need to indicate the position of the size standards on their DNA gels shown in Figure 2C.

c. The authors need to indicate the position of their molecular weight markers on their Western blots shown in Figure 2D.

d. The authors need to provide some quantitative analysis of the histomorphology results shown in Figure 2I as well as the acrosome and nucleus morphology results shown in Figure 2J. Without this quantification, it is difficult to see how the inactivation of Centlein impacts spermatogenesis based on the images provided in these figures.

9) Figure 3.

a. The authors need to provide some quantitative analysis of the H&E staining results shown in Figure 3A.

b. The authors need to provide some quantitative analysis of the TEM results shown in Figure 3E. How often do they see these ultrastructural defects in Centlein-/- sperm? Is the missing microtubule doublet phenotype observed throughout the entire length of the Centlein-/- sperm axoneme?

10) Figure 4.

a. The authors need to provide some quantitative analysis of the PAS and hematoxylin staining results shown in Figure 4B.

b. The authors need to provide some quantitative analysis of the TEM results shown in Figure 4C.

11) Figure 5.

a. The authors need to indicate the position of their molecular weight markers on their Western blots shown in Figures 5A-B.

b. The authors need to provide some quantitative analysis of the immunofluorescence results shown in Figures 5C-D. For example, what is the extent of the co-localization of CEP135 and CENTLEIN over the course of spermatogenesis? What is the extent of the co-localization of SUN5 and CENTLEIN or PMFBP1 and CENTLEIN?

c. Why does SUN5 show such a punctate localization in the sperm nuclear envelope? Is this a result of the particular optical section shown in Figure 5D? In addition, what are the clumps of PMFBP1 that do not co-localize with CENTLEIN in Figure 5E? How specific are the antibodies for SUN5 and PMFBP1? Do the authors see the same subcellular localization of SUN5 and PMFBP1 with other antibodies for these proteins?

d. The authors might consider showing a maximum intensity projection instead of a single optical section.

12) Figure 6.

a. The authors need to provide some quantitative analysis of the immunofluorescence data shown in Figures 6A-D.

b. Why is the CENTLEIN signal in Figure 6A so weak relative to the other CENTLEIN images shown in this Figure?

c. The images of SUN5 provided in Figure 6B make me seriously question the specificity of the anti-SUN5 antibody used in this work. Given this concern, I find it difficult to be able to agree with the authors' conclusion that "Ablation of Centlein impairs the localization of PMFBP1 to the coupling apparatus.

d. Also regarding Figure 6D, what are the non-PMFBP1-associated SUN5 clumps? Also, why does

the extranuclear SUN5 signal increase in the Centlein-/- testicular germ cells relative to controls? The authors need to discuss this strange result.

Minor issues.

- 1) All abbreviations used in the manuscript need to be defined.
- 2) In the Antibodies section of the Methods, the authors need to provide the antibody dilutions used to generate the data presented in this work.
- 3) In the Epididymal Sperm Count section of the Methods, the authors need to provide more information regarding the microscopes used to generate the data presented in this work. Specifically, they need to report which objectives (magnification, correction level, N.A.), detectors, light sources, software, excitation filters, and emission filters were used.
- 4) In the Transmission electron microscopy section of the Methods, the authors completely neglect to provide any information about the microscope they used to generate the TEM data presented in this work. This must be rectified.
- 5) Pages 7-8, lines 163-164: The statement "amount of sperm tail could be stained by eosin in Centlein -/- caudal epididymis" does not make sense and needs to be clarified.
- 6) Page 9, lines 199-200: The authors state, "...the flagellum was always tightly attached to the nuclear envelope". The word "tightly" is misleading here, as the strength of the interaction was not measured. How are the authors determining the tightness of this interaction?
- 7) Page 9, lines 202-203: How do the authors know that they are seeing "the destroyed coupling apparatus scattered in the cytosol of the elongated spermatid" in Figure 4C? Did they perform immunogold EM?
- 8) Page 9, line 207: The authors state, "Inconceivably, these severe defects resulting from the lack of CENTLEIN broke the sperm head-tail junction during sperma(t)ogenesis". However, "Inconceivably" seems to be inappropriately used here.
- 9) Pages 9-10, lines 208-214: The authors suggest that CENTLEIN may be performing two possible roles in the HTCA. However, the language used to describe these roles makes it seem like it is known that CENTLEIN works in these ways. The authors should use conditional language here rather than definitive language.
- 10) Page 13, line 280: The statement "identified as the first centrosome protein" does not make sense here. Was CENTEIN the first centrosome protein identified?

Reviewer #2:

Remarks to the Author:

The paper "The missing linker between SUN5 and PMFBP1 in sperm head-tail coupling apparatus" by Ying Zhang and Li Yuan lab is fascinating. The paper address one of the critical questions in sperm biology - how the sperm head connects to the tail, a problem that has made significant progress in the lads few years. This paper shows that CENTLEIN is a new and essential component of this connecting mechanism that is bridging between 2 others already known connecting proteins, SUN5, and PMFBP1. This paper is a continuation of past published work by Li Yuan lab, demonstrating that CENTLEIN is a centrosomal protein that is localized at the proximal ends of centrioles and is required for centrosome cohesion.

The paper is very promising, and I have enjoyed reading it, but it needs to be better developed. Based on my comment below, I think this can be done quite quickly by the Li Yuan lab.

Major point

- CENTLEIN's localization within the sperm neck in the spermatid and spermatozoon is not precisely determined - is it in the distal centriole? Proximal centriole" both? None? This is critical to the interpretation of the paper.
- Fig. 5c and Fig 6a-d - provide an inset with a zoon on the 2 centrioles - it is hard to see the colocalization. These tow figures need internal annotation - it is not clear what do we see in the figures.
- Fig 6b - Sun 5 immunostaining is not convincing - please take this data from the paper or better test it.
- "whereas PMFBP1 could not attach to the implantation fossa of the sperm nucleus (Fig. 6b)". Figures 6a, b, c, and d are unclear. You need to add a phase picture or some counter stain so we can see the boundary of the sperm head and tail in all panels. Clear images of heads and tails need to be present. Quantification needs to be performed. Insets with enlarged head-neck and

tail-neck need to be added. As presented, with this quality of data, I am not convinced in SUN5-CENTLEIN-PMFBP1.

Minor points

Introduction

- "During development of the sperm tail there is also a distal centriole at the base of the axoneme oriented approximately at a right angle to the proximal centriole." The distal centriole is also present in the mature centriole – not only developmentally (Fishman et al., 2018 - A novel atypical sperm centriole is functional during human fertilization. Nature Communications.). Please correct and add the citation.

Figures

- Fig 1a – Add molecular weight marker and mark the location of the screened protein so they can be identified and evaluated relative to the other bands. These need to be added to all other western figures in the paper.

- Fig 1b-g – why sometimes Lane 3 of these figures show a protein – what is this protein?

- Fig 3c – the figure seems to show that the tail is shorter in the mutant – please quantify.

- Fig6 – the stage of the sperm in the figure needs to be better define. "germ cells" is not sufficiently clear.

Results

- "In addition, the axoneme of decapitated tails in Centlein^{-/-} mice was also 172 perturbed (Fig. 3e). – please quantify. – please quantify.

- "We detected the proportion of decapitated tails in the Centlein^{-/-} corpus and caput 181 of the epididymis, and found they were similar to those in the Centlein^{-/-} caudal 182 epididymis (Fig. 3d and 4a). Thus, the detachment of the sperm head and tail in 183 Centlein^{-/-} mice may occur within the seminiferous tubules." Or earlier in the epididymis.

- Although the acrosome biogenesis and the process of the 186 sperm head shaping showed normal in Centlein^{-/-} mice (Fig. 2j and 4b). - please quantify the number of heads per surfers unit.

- "which might be 189 caused by the sperm head detachment from flagellum during spermiogenesis (Fig. 4b)." It is not clear to me how you see sperm head detachment from flagellum in Fig. 4b.

- "while mature spermatozoa were released into the lumen of the 192 seminiferous tubules at stage VIII in Centlein^{+/+} testes (Fig. 4b)," where is the lumen in Fig. 4b?

- "In 196 Centlein^{+/+} step7-9 spermatids, the ultrastructure of the HTCA was fully assembled 197 and consisted of a well-defined segmented column (Sc), capitulum (Cp), and basal 198 plate (Bp) (Fig. 4c)." The 2 centrioles are a critical component of the HTCA – what happened to them? Why they are not both marked in the Fig. 4c on the mutant side?

- "two types of HTCA ultrastructure were detected 201 in Centlein^{-/-} mice" what is their frequency? What happened to the centriole in each one of them? Fore claret- draw a picture describing each phenotype.

Discussion

- Since CENTLEIN is a known centrosome protein there, needs to be a discussion of its role in the mice sperm that is thought not to have a centrosome (please see the paper "It Takes Two (Centrioles) to Tango"). This is important in light of the recent discovery that the human sperm centrioles are atypical (Fishman et al., 2018 - A novel atypical sperm centriole is functional during human fertilization. Nature Communications.).

- "CENTLEIN, identified as the first centrosome protein, is essential for the integrity of the HTCA (Fig. 4)." What about this paper "Rat hd Mutation Reveals an Essential Role of Centrobin in Spermatid Head Shaping and Assembly of the Head-Tail Coupling Apparatus"

Sincerely,

Tomer Avidopr-Reiss

We thank the two referees for close scrutiny of the manuscript and we are grateful for the opportunity to resubmit a revised version of the manuscript. We have now completed an extensive revision of the manuscript and addressed the referee's concerns.

REVIEWER COMMENTS

Reviewer #1 (Remarks to the Author):

This work investigates how SUN5 and PMFBP1 function together in the head-to-tail coupling apparatus (HTCA) during spermatogenesis. This information is important, as mutations in the genes that encode SUN5 and PMFBP1 were previously associated with acephalic spermatozoa syndrome. The authors provide evidence to support the hypothesis that SUN5 and PMFBP1 indirectly interact with each other through the centrosomal protein CENTLEIN. They show that male mice lacking CENTLEIN are sterile due to their production of acephalic sperm. Transmission electron microscope-based analysis of the ultrastructure of sperm produced by CENTLEIN-null mice revealed defects in the assembly of the segmented column and the capitulum of the HTCA at early stages of spermatogenesis as well as the detachment of the anomalous segmented column and basal plate from spermatid nuclear envelopes. Based on confocal images of immunofluorescently stained sperm isolated from CENTLEIN-, PMFBP1-, or SUN5-null mice, the authors conclude that CENTLEIN/SUN5/PMFBP1 function during HTCA assembly and the resulting integration of the sperm head and tail. These results suggest that mutations in CENTLEIN may also be associated with acephalic spermatozoa syndrome in humans. Overall, I feel that this manuscript is well written as well as tackles an interesting and important question in the field of spermatogenesis. However, I am not convinced that the data presented in this manuscript fully supports the authors' conclusions. Moreover, I do not feel that this work represents enough of a mechanistic advance for the field. Thus, I cannot recommend that this manuscript be accepted for publication in its current state. Below, I will briefly outline the major and minor issues that I feel that the authors must address in order for their manuscript to be accepted for publication.

Major issues.

1) The authors need to better describe how they envision that SUN5 functions in the HTCA. SUN5 is a member of the SUN family of inner nuclear membrane proteins, yet the authors fail to mention this important fact anywhere in their manuscript. Nor do they mention that SUN proteins are core components of the linker of nucleoskeleton and cytoskeleton complex due to their ability to form a transluminal interaction with the KASH/nesprin family of outer nuclear membrane proteins. Based on this information, do the authors propose that the normally nucleoplasmic N-terminus of SUN5 resides on the outer nuclear membrane where it is able to interact with CENTLEIN? Alternatively, does SUN5 function to recruit a KASH/nesprin protein to

the outer nuclear membrane of the sperm nucleus that is capable of interacting with CENTLEIN? The authors need to discuss these opposing models in their manuscript.

These are the very important and insightful comments brought forward by Reviewer #1, for which we are very grateful. It has been reported that the overexpressed SUN5 in somatic cells is localized at the inner nuclear membrane¹. However, few work has addressed the precise localization of SUN5 in sperm nuclear envelope, as the complicated localization of SUN5 in different studies^{1,2}. Previously, we found SUN5 was first expressed in the nuclear envelope and later migrated to the coupling apparatus of the sperm during sperm head elongation and differentiation³. Furthermore, the SUN5C (coiled-coil domain and the SUN domain of SUN5) region of SUN5 does not interact with KASH5 LR⁴, indicating SUN5 might be distinct from the classical SUN family proteins. By using LC-MS analysis, we found SUN5 could directly bind to the type II heat shock protein 40 protein, DNAJB13, which might facilitate SUN5 protein folding to ensure the interaction between the implantation fossa and unknown protein in the HTCA⁴. Therefore, SUN5 might directly bind some cytoplasm proteins to achieve its physiological function during spermatogenesis. To further explore the underlying mechanism of SUN5 recruitment of CENTLEIN, we firstly mapped the CENTLEIN binding region in SUN5 according Reviewer #1's suggestion, and found the SUN domain of SUN5 was sufficient to bind CENTLEIN (Fig. 6c). In addition, the GST-pulldown experiments showed the SUN domain of SUN5 could directly interact with the 971–1396aa region of CENTLEIN (Fig. 6e). The immunofluorescent staining also showed CENTLEIN could partially colocalize with SUN5 in the HTCA (Fig. 5e). Thus, SUN5 might directly recruit CENTLEIN through its SUN domain during spermatogenesis. Many thanks for helping us clarify this important issue, and we have introduced the cellular functions of SUN5 in the introduction part and discussed the potential model of SUN5 binding CENTLEIN in the discussion as below.

“SUN5 is a transmembrane protein located on the nuclear envelope, which could interact with the coupling apparatus-related protein DnaJ heat shock protein family (Hsp40) member B13 (DNAJB13) to facilitate SUN5 protein folding to ensure the interaction between the implantation fossa and an unknown protein in the HTCA.”

“SUN5 belongs to the SUN domain protein in the inner nuclear membrane, which could fasten the linkage between the nuclear lamina and cytoskeleton through the SUN-KASH interfaces^{1,5}. Although the overexpressed SUN5 in somatic cells is localized at the inner nuclear membrane¹, few work has addressed the precise localization of SUN5 in sperm nuclear envelope, as the complicated localization of SUN5 in different studies^{1,2}. Previously, we found the SUN5C (coiled-coil domain and the SUN domain of SUN5) region of SUN5 does not interact with KASH5 LR⁴, indicating SUN5 might be distinct from the classical SUN domain proteins. By using LC-MS analysis, we found SUN5 could directly bind to the DNAJB13 to facilitate SUN5 protein folding, which is required for the integrity of the HTCA⁴. Here, we found the SUN domain of SUN5 could directly interact with the 971–1396aa region of CENTLEIN (Fig. 6e). The

immunofluorescent staining showed CENTLEIN could partially colocalize with SUN5 in the HTCA (Fig. 5e), and absence of SUN5 might perturb the HTCA localization of CENTLEIN (Fig. 7 i, j). Thus, SUN5 might directly recruit CENTLEIN to the HTCA through its SUN domain."

2) The authors need to provide of what is known about the cellular functions of the SPATA6 and PMFBP1 proteins. Without this background information, it is difficult to understand how these proteins might be working together with SUN5 and CENTLEIN in the HTCA.

Thank you for this kindly suggestion, and we have added a brief description for SPATA6 and PMFBP1 in the introduction part as below:

"As SPATA6 could interact with myosin light-chain polypeptide 6 (MYL6) on the manchettes of elongating spermatids, it might be involved in myosin-based microfilament transport for the assembly of the segmented columns and capitulum during sperm head-tail coupling apparatus (HTCA) formation."

"PMFBP1 was firstly identified as a polyamine modulating factor 1 (PMF1) binding protein, which could enhance the catabolism and recycling of polyamines⁶. During spermatogenesis, PMFBP1 is specifically expressed in adult testis and predominantly located in the HTCA."

3) Continuing along the theme of mechanism of SUN5 function described above, the authors really need to map the CENTLEIN binding site in SUN5. They also should map the SUN5 and PMFBP1 binding sites on CENTLEIN. Moreover, they need to test if the SUN5-CENTLEIN interaction were direct or not using recombinant proteins. Despite what the authors state in line 95 of the Introduction, the ability of two proteins to be co-immunoprecipitated does not prove that they are able to directly interact with each other. While I understand that the pandemic makes it difficult to perform additional experiments at this time, I strongly feel that a deeper mechanistic understanding of the CENTLEIN/SUN5/PMFBP1 interaction is needed for this manuscript to be suitable for publication at Nat Commun.

We thank the reviewer for the constructive comments.

(1) Deletion analysis of CENTLEIN showed that aa 971-1396 was sufficient to bind SUN5 (Fig. 6a), and 601-1396aa was required for its binding to PMFBP1 (Fig. 6b). Two portions of PMFBP1 encompassing residues 1-282 and 750-1023 were necessary for its binding to CENTLEIN (Fig. 6d), while SUN domain of SUN5 was sufficient to bind CENTLEIN (Fig. 6c).

(2) An *in vitro* direct binding assay was subsequently conducted. GST-SUN domain of SUN5 readily pulldowned CENTLEIN 971-1396aa (Fig. 6e), and recombinant PMFBP1 750-1023aa could directly interact with CENTLEIN 601-970aa (Fig. 6f).

(3) We have now shown that CENTLEIN mediates an interaction between SUN5 and PMFBP1 (Fig. 6g).

This evidence, along with a lack of interaction between SUN5 and PMFBP1,⁷ suggests that SUN5 complexes with PMFBP1 through the independent direct interactions of both proteins with CENTLEIN (Fig.6).

4) The authors fail to mention the source of the cDNA constructs that encode the EGFP-tagged centrosome proteins used in Figure 1. Are these previously described constructs? Are they functional? If they are not, the authors should provide representative images of the subcellular localization of these EGFP-tagged centrosome proteins in the HEK293T cells to demonstrate that these constructs are properly localized to the centrosome.

We sincerely apologize for not mentioning the source of the cDNA constructs that encode the EGFP-tagged centrosome proteins used in Figure 1. Now, we have provided the information of these cDNA constructs in Methods. Centrosome localization of these EGFP-tagged proteins are now presented in Supplemental Figure 1. P50 is a subunit of the dynactin complex localized at the centrosome^{8,9,10}, and its overexpression has been used to study dynactin-dynein-related centrosome proteins, albeit without clear centrosome localization^{8,9,10}.

5) The authors should provide a better description of the different stages of spermatogenesis for their readers who may not be experts. Without this information, the manuscript is difficult to follow.

We thank the reviewer for this invaluable suggestion. A better description of the different stages of spermatogenesis has now been added.

“Spermatogenesis is the process of haploid male gamete production with successive cellular differentiation^{11,12}. During spermatogenesis, germ cells undergo meiosis to ensure haploidization of the genome and genetic diversity^{13,14}, and the haploid germ cells subsequently undergo a dramatic morphological change and nuclear chromatin re-organization to form spermatozoon, during this process, the cytoplasm needs to be removed, and forming two specific structures termed as acrosome and flagellum^{14,15}.”

6) Page 11, lines 245-247: The authors state “PMFBP1 could not attach to the implantation fossa of the sperm nucleus...Thus CENTLEIN is essential for the connection between SUN5 and PMFBP1 in the HTCA”. However, they do not provide enough experimental evidence to support these claims. The authors do not know if PMFBP1 cannot interact with the implantation fossa of the sperm nucleus nor do they know that CENTLEIN is essential for the SUN5-PMFBP1 interaction.

We greatly appreciate the reviewer’s comments. We have now realized that our data were not sufficient to support this statement. As SUN5 has been reported to

be localized on the coupling apparatus of the sperm head and tail ³, and they falling apart in *Centlein*^{-/-} spermatids, we have toned down this claim in the re-drafted manuscript (as below):

“PMFBP1 could not attach to the head-tail coupling apparatus.”

To further confirm CENTLEIN essential for the SUN5-PMFBP1 interaction, we performed the Co-IP in HEK293T cells transfected with a GFP-tagged PMFBP1, FLAG-tagged SUN5 and MYC-tagged CENTLEIN (Fig. 6g). We found SUN5 could interact with PMFBP1 when CENTLEIN was presented (Fig. 6g).

7) Figure 1.

a. It appears that FLAG-SUN5 can also interact with the following GFP-tagged centrosome proteins: BBS4m, NEK2, CPAP, and P50. While these interactions are clearly less pronounced than what was observed with CENTLEIN, they are completely ignored in the manuscript. Are these 4 proteins known to be involved in spermatogenesis? This needs to be discussed.

We thank the reviewer for pointing this out. As the signal of BBS4, NEK2, CPAP, and P50 in Fig. 1a was similar with the negative control, they should be caused by nonspecific binding on the protein-G beads. We have redone the experiment(s) (Fig. 1a) with more washing times using ELB buffer, and found SUN5 could only bind to CENTLEIN (Fig. 1a).

b. The authors need to indicate the position of their molecular weight markers on their Western blots.

We fully agreed and have made the change accordingly.

c. The authors should use asterisks to label the bands that they want their readers to focus on in the GFP blot shown in Figure 1A.

We fully agreed and have made the change accordingly (Fig. 1a).

d. What are the big black bands found in the FLAG blot shown in Figure 1G? Are they IgG light or heavy chains? The amount of over-saturation of these big black bands makes it difficult to see the immunoprecipitated FLAG-SPATA6 band.

Yes, they are IgG heavy chain(s).

8) Figure 2.

a. The authors need to indicate the 5' and 3' ends of the DNA sequences shown in Figure 2A.

We fully agreed and have made the change accordingly (Supplemental Figure 2b).

b. The authors need to indicate the position of the size standards on their DNA gels shown in Figure 2C.

We fully agreed and have made the change accordingly (Supplemental Figure 2c).

c. The authors need to indicate the position of their molecular weight markers on their Western blots shown in Figure 2D.

We fully agreed and have made the change accordingly (Fig. 2a).

d. The authors need to provide some quantitative analysis of the histomorphology results shown in Figure 2I as well as the acrosome and nucleus morphology results shown in Figure 2J. Without this quantification, it is difficult to see how the inactivation of Centlein impacts spermatogenesis based on the images provided in these figures.

We fully agreed. Now, the quantitative analysis of the histomorphology results has been provided (Fig. 2g, i).

9) Figure 3.

a. The authors need to provide some quantitative analysis of the H&E staining results shown in Figure 3A.

The H&E staining in Figure 3a showed apparent absence of sperm heads in the *Centlein*^{-/-} epididymis, as the spermatozoa appeared to be stained less with hematoxylin in the *Centlein*^{-/-} mice. The precise quantification of the acephalic spermatozoa in *Centlein*^{-/-} mice has been shown in Fig. 3d.

b. The authors need to provide some quantitative analysis of the TEM results shown in Figure 3E. How often do they see these ultrastructural defects in *Centlein*^{-/-} sperm? Is the missing microtubule doublet phenotype observed throughout the entire length of the *Centlein*^{-/-} sperm axoneme?

Quantitative analysis of the ultrastructural defects in the mid-piece, principal piece and end piece of *Centlein*^{-/-} spermatozoa has been presented (Fig.3 f-i), respectively. The precise quantification of the acephalic spermatozoa in *Centlein*^{-/-} caudal epididymis has been shown in Fig. 3d.

10) Figure 4.

a. The authors need to provide some quantitative analysis of the PAS and hematoxylin staining results shown in Figure 4B.

Quantitative analysis of the PAS results has now been added (please see Fig. 4c).

b. The authors need to provide some quantitative analysis of the TEM results shown in Figure 4C.

As a single cell could be cut into different sections, it might be captured repeatedly in TEM experiment, thus it was difficult to quantify the broken sperm head-tail junction by TEM. We have collected all TEM sections and tried to quantify the two types of defective HTCA ultrastructure in *Centlein*^{-/-} spermatids, i.e. the detached coupling apparatus from the spermatid nucleus and unassembled or destroyed coupling apparatus, has now been provided, respectively (Fig. R1).

Figure R1

Figure R1 Quantitative analysis of the two types of defective HTCA ultrastructure in *Centlein*^{+/+} and *Centlein*^{-/-} step 10-12 spermatids. Data are presented as mean \pm SEM. * $P < 0.05$.

11) Figure 5.

a. The authors need to indicate the position of their molecular weight markers on their Western blots shown in Figures 5A-B.

We fully agreed and have made the change accordingly (Fig. 5a, b).

b. The authors need to provide some quantitative analysis of the immunofluorescence results shown in Figures 5C-D. For example, what is the extent of the co-localization of CEP135 and CENTLEIN over the course of spermatogenesis? What is the extent of the co-localization of SUN5 and CENTLEIN or PMFBP1 and CENTLEIN?

This is a very important point brought forward by the referee.

(1) Using Super-Resolution microscopy and an antibody against CETN1/2, a standard marker for centrioles in mammalian spermatids, we found that CENTLEIN localized to both distal and proximal centrioles of round, elongating, and early elongated spermatids (Fig. 5d).

(2) In somatic cells, CETN1/2 is a marker of the distal ends of centrioles, whereas Cep135 is a proximal centriolar marker complexed with both C-Nap1 and CENTLEIN^{16, 17}, and they partially co-localized with CENTLEIN in spermatids (Fig.5c). The extent of the co-localization of CEP135 and CENTLEIN has been provided in Fig. 5c.

(3) Both SUN5 and PMFBP1 are confined to the coupling apparatus stained as a dot at the late stage of spermiogenesis, whereas CENTLEIN staining vanishes at Step15 spermatids and onwards.

(4) Co-localization between SUN5 and CENTLEIN, as well as PMFBP1 and CENTLEIN occurs during Step 10-14. The extent of the co-localization of SUN5 and CENTLEIN or PMFBP1 and CENTLEIN has been provided in Fig. 5e and 5f.

c. Why does SUN5 show such a punctate localization in the sperm nuclear envelope? Is this a result of the particular optical section shown in Figure 5D? In addition, what are the clumps of PMFBP1 that do not co-localize with CENTLEIN in Figure 5E? How specific are the antibodies for SUN5 and PMFBP1? Do the authors see the same subcellular localization of SUN5 and PMFBP1 with other antibodies for these proteins?

We thank the reviewer for questioning the specificity of the anti-SUN5 and anti-PMFBP1 antibodies used in the previous manuscript. The new batches of the anti-SUN5 and anti-PMFBP1 antibodies have been employed in the revised manuscript (Fig. 5e, f).

d. The authors might consider showing a maximum intensity projection instead of a single optical section.

As suggested, we now used a maximum intensity projection instead of a single optical section (Fig. 5c,d).

12) Figure 6.

a. The authors need to provide some quantitative analysis of the immunofluorescence data shown in Figures 6A-D.

Quantitative analysis of the immunofluorescence data has been added (please see Fig. 7b, c, e, f, h, j).

b. Why is the CENTLEIN signal in Figure 6A so weak relative to the other CENTLEIN images shown in this Figure?

We thank the reviewer for pointing this out. We have redone the experiment(s) (Supplementary Fig. 5).

c. The images of SUN5 provided in Figure 6B make me seriously question the specificity of the anti-SUN5 antibody used in this work. Given this concern, I find it difficult to be able to agree with the authors' conclusion that "Ablation of Centlein impairs the localization of PMFBP1 to the coupling apparatus."

We fully agreed. The SUN5-specific antibody has been used in the revised manuscript (Fig. 7d, g).

d. Also regarding Figure 6D, what are the non-PMFBP1-associated SUN5 clumps? Also, why does the extranuclear SUN5 signal increase in the Centlein-/- testicular germ cells relative to controls? The authors need to discuss this strange result.

We sincerely apologize for presenting the blurry figure, which has now been rectified (Fig. 7d, g).

Minor issues.

1) All abbreviations used in the manuscript need to be defined.

We fully agreed and have added all abbreviations used in the revised manuscript.

2) In the Antibodies section of the Methods, the authors need to provide the antibody dilutions used to generate the data presented in this work.

The antibody dilutions have been provided in the revised manuscript.

3) In the Epididymal Sperm Count section of the Methods, the authors need to provide more information regarding the microscopes used to generate the data presented in this work. Specifically, they need to report which objectives (magnification, correction level, N.A.), detectors, light sources, software, excitation filters, and emission filters were used.

The detail information regarding the microscopes has been provided in the revised manuscript.

4) In the Transmission electron microscopy section of the Methods, the authors completely neglect to provide any information about the microscope they used to generate the TEM data presented in this work. This must be rectified.

We thank the reviewer for pointing this out. This has been rectified.

5) Pages 7-8, lines 163-164: The statement “amount of sperm tail could be stained by eosin in Centlein ^{-/-} caudal epididymis” does not make sense and needs to be clarified.

We fully agreed and have restated this in the revised version of the manuscript (as below).

“Although the spermatozoa in the Centlein^{-/-} caudal epididymis appeared to be stained less with hematoxylin compared with those in the Centlein^{+/-} and Centlein^{+/+} caudal epididymis (Fig. 3a), eosin staining in Centlein^{-/-} caudal epididymis was indistinguishable from the control groups, suggesting that sperm heads in the Centlein^{-/-} epididymis are much less abundant.”

6) Page 9, lines 199-200: The authors state, “...the flagellum was always tightly attached to the nuclear envelope”. The word “tightly” is misleading here, as the strength of the interaction was not measured. How are the authors determining the tightness of this interaction?

We thank the reviewer for pointing this out. This has been rectified.

“Having spermatids elongated, the Centlein^{+/+} spermatid coupling apparatus together with the flagellum was always attached to the nuclear envelope (Fig. 4e).”

7) Page 9, lines 202-203: How do the authors know that they are seeing “the destroyed coupling apparatus scattered in the cytosol of the elongated spermatid” in Figure 4C? Did they perform immunogold EM?

We thank the reviewer for pointing this out. As the ultrastructure of the HTCA has been well described, it could be characterized by the segmented column (Sc), capitulum (Cp), basal plate and centrioles. Although lacking of immunogold EM, the Sc, Bp and PC could be characterized in Fig. 4e. On the other hand, because of the technique difficulty and lacking high quality of any coupling apparatus related antibodies, such as SUN5, PMFBP1, it was extremely difficult to perform the immunogold electron microscopical study currently.

8) Page 9, line 207: The authors state, “Inconceivably, these severe defects resulting from the lack of CENTLEIN broke the sperm head-tail junction during sperma(t)ogenesis”. However, “Inconceivably” seems to be inappropriately used here.

“Inconceivably” has been removed.

9) Pages 9-10, lines 208-214: The authors suggest that CENTLEIN may be performing two possible roles in the HTCA. However, the language used to describe

these roles makes it seem like it is known that CENTLEIN works in these ways. The authors should use conditional language here rather than definitive language.

We fully agreed and have restated in the revised version of the manuscript (as below).

“These results suggest that CENTLEIN might play two functional roles in HTCA: first, CENTLEIN might work as a bona fide centrosomal component initiating assembly of the ultra-structural components of the HTAC; second, CENTLEIN might be required for the tight attachment of the coupling apparatus to the caudal portion of the sperm head.”

10) Page 13, line 280: The statement “identified as the first centrosome protein” does not make sense here. Was CENTEIN the first centrosome protein identified?

We sincerely apologize for the overstatement. This has now been rectified.

“Here, we show the centrosome protein, CENTLEIN, is essential for the integrity of the HTCA (Fig. 4f).”

Reviewer #2 (Remarks to the Author):

The paper “The missing linker between SUN5 and PMFBP1 in sperm head-tail coupling apparatus” by Ying Zhang and Li Yuan lab is fascinating. The paper address one of the critical questions in sperm biology - how the sperm head connects to the tail, a problem that has made significant progress in the lads few years. This paper shows that CENTLEIN is a new and essential component of this connecting mechanism that is bridging between 2 others already known connecting proteins, SUN5, and PMFBP1. This paper is a continuation of past published work by Li Yuan lab, demonstrating that CENTLEIN is a centrosomal protein that is localized at the proximal ends of centrioles and is required for centrosome cohesion.

The paper is very promising, and I have enjoyed reading it, but it needs to be better developed. Based on my comment below, I think this can be done quite quickly by the Li Yuan lab.

Major point

- CENTLEIN’s localization within the sperm neck in the spermatid and spermatozoon is not precisely determined – is it in the distal centriole? Proximal centriole" both? None? This is critical to the interpretation of the paper.

Thanks to Fishman et al’s work,¹⁸ we are able to employ a modified protocol to detect the detail localization of CENTLEIN within the sperm neck of the spermatids. Prof. Avidopr-Reiss’s concerns are now addressed explicitly.

(1) CENTLEIN is expressed as a 180-kDa protein in testicular spermatogenic cells and testicular sperm, but cannot be detected in epididymal sperm. CENTLEIN is localized in the coupling apparatus of the round, elongating, and early elongated spermatids, while it disappears at Step15&16 of spermatids.

(2) Using Super-Resolution microscopy and an antibody against CETN1/2, a standard marker for centrioles in mammalian spermatids¹⁸, we have demonstrated that CENTLEIN localizes to both distal and proximal centrioles of round, elongating, and early elongated spermatids (Fig. 5d).

- Fig. 5c and Fig 6a-d – provide an inset with a zoom on the 2 centrioles – it is hard to see the colocalization. These two figures need internal annotation – it is not clear what do we see in the figures.

We sincerely apologize for presenting the unclear pictures. As suggested, we now zoom in the 2 centrioles in Fig. 5 and Fig. 7, which are annotated internally.

- Fig 6b – Sun 5 immunostaining is not convincing – please take this data from the paper or better test it.

We fully agreed. A new batch of the SUN5-specific antibody has been used in the revised manuscript (Fig. 7d, g).

- “whereas PMFBP1 could not attach to the implantation fossa of the sperm nucleus (Fig. 6b)”. Figures 6a, b, c, and d are unclear. You need to add a phase picture or some counter stain so we can see the boundary of the sperm head and tail in all panels. Clear images of heads and tails need to be present. Quantification needs to be performed. Insets with enlarged head-neck and tail-neck need to be added. As presented, with this quality of data, I am not convinced in SUN5-CENTLEIN-PMFBP1.

We sincerely apologize for presenting an unqualified figure. As suggested, the neck regions of the spermatids are now demarcated and zoomed in (Fig. 5 and Fig. 7). Quantitative analysis of the immunofluorescence data was also presented (please see Fig. 7b, c, e, f, h, j).

Minor points

Introduction

- “During development of the sperm tail there is also a distal centriole at the base of the axoneme oriented approximately at a right angle to the proximal centriole.” The distal centriole is also present in the mature centriole – not only developmentally (Fishman et al., 2018 - A novel atypical sperm centriole is functional during human fertilization. Nature Communications.). Please correct and add the citation.

We sincerely apologize for missing the important publication. We have now rewritten and cited this important paper.

Figures

- Fig 1a – Add molecular weight marker and mark the location of the screened protein so they can be identified and evaluated relative to the other bands. These need to be added to all other western figures in the paper.

We fully agreed and have made the change accordingly.

- Fig 1b-g – why sometimes Lane 3 of these figures show a protein – what is this protein?

We thank Prof. Avidopr-Reiss for pointing this out. We have redone the experiment(s) (Fig. 1b-g), and these bands might be some nonspecific signal, which might be caused by IgG (Supplementary Fig. 6).

- Fig 3c – the figure seems to show that the tail is shorter in the mutant – please quantify.

As suggested, the length of the epididymal sperm tails has been quantitated (Fig. R2), and we found the Centlein-null sperm tail was indeed shorter than that of control group.

Figure R2

Figure R2 The disruption of Centlein impaired the sperm tail length.

(a) Single-sperm staining was performed using *Centlein^{+/+}* and *Centlein^{-/-}* spermatozoa. Nuclei were stained with DAPI (blue).

(b) Quantitative analysis of sperm tail length in *Centlein^{+/+}* and *Centlein^{-/-}* spermatozoa. Data are presented as mean \pm SD. * $P < 0.05$.

- Fig6 – the stage of the sperm in the figure needs to be better define. "germ cells" is not sufficiently clear.

We fully agreed and have made the change accordingly (Fig. 7).

Results

- "In addition, the axoneme of decapitated tails in *Centlein^{-/-}* mice was also 172 perturbed (Fig. 3e). – please quantify. - please quantify.

Quantitative analysis of the ultrastructural defects in the mid-piece, principal piece and end piece of *Centlein^{-/-}* sperm has been presented (Fig. 3f-i), respectively.

- "We detected the proportion of decapitated tails in the *Centlein^{-/-}* corpus and caput 181 of the epididymis, and found they were similar to those in the *Centlein^{-/-}* caudal 182 epididymis (Fig. 3d and 4a). Thus, the detachment of the sperm head and tail in 183 *Centlein^{-/-}* mice may occur within the seminiferous tubules." Or earlier in the epididymis.

We thank Prof. Avidopr-Reiss for pointing this out. This has been rectified.

*"Thus, the detachment of the sperm head and tail in *Centlein^{-/-}* mice may occur within the seminiferous tubules or entrance into the caput of the epididymis."*

- Although the acrosome biogenesis and the process of the 186 sperm head shaping showed normal in *Centlein^{-/-}* mice (Fig. 2j and 4b). - please quantify the number of heads per surfers unit.

Quantitative analysis of germ cells per surfer unit has been presented (Fig. R3), and we found germ cells showed a little decrease in step 9-12 and step16 in *Centlein^{-/-}* mice.

Figure R3

Figure R3 Quantitative analysis of germ cells per surfer unit in *Centlein*^{+/+} and *Centlein*^{-/-} mice. Data are presented as mean ± SEM. **P* < 0.05.

- “which might be 189 caused by the sperm head detachment from flagellum during spermiogenesis (Fig. 4b).” It is not clear to me how you see sperm head detachment from flagellum in Fig. 4b.

We thank Prof. Avidopr-Reiss for pointing it out.

In the WT testis, the well-shaped spermatozoa had migrated to the edge of the seminiferous epithelium with their head and acrosomic system oriented toward the basement membrane (Fig. 4d, e).

In *Centlein*^{-/-} testes, it was quite different; although the shape of sperm head was normal, they were not oriented toward the basement membrane, as most of their heads were oriented toward the lumen of the seminiferous tubules (Fig. 4d, e).

Stage VII-VIII is the so-called spermiation phase when mature spermatozoa are ready to be released. This indicates that the *Centlein*-null sperm head and tail might break apart during spermiation so that the separated sperm head cannot align itself in the right orientation.

- “while mature spermatozoa were released into the lumen of the 192 seminiferous tubules at stage VIII in *Centlein*^{+/+} testes (Fig. 4b),” where is the lumen in Fig. 4b?

We sincerely apologize for presenting the unclear pictures, and we have labeled the lumen and basement membrane in Fig. 4b, d, e.

- "In 196 *Centlein*^{+/+} step 7-9 spermatids, the ultrastructure of the HTCA was fully assembled 197 and consisted of a well-defined segmented column (Sc), capitulum (Cp), and basal 198 plate (Bp) (Fig. 4c)." The 2 centrioles are a critical component of the HTCA – what happened to them? Why they are not both marked in the Fig. 4c on the mutant side?

We sincerely apologize for not pointing the centrioles out, and we have now labeled the centrioles in some cells (Fig. 4f), but not all, most likely due to failure in simultaneous acquirement of both PC and DC in the same section. Alternatively, we labeled the spermatids with an antibody against CETN1/2 reported to be localized on both PC and DC of the spermatids¹⁸, and detected two dots in a *Centlein*^{-/-} spermatid, indicating that the *Centlein*-null spermatid may still contain both PC and DC. However, we do not know whether *Centlein*^{-/-} PC and or DC remain intact.

- "two types of HTCA ultrastructure were detected 201 in *Centlein*^{-/-} mice" what is their frequency? What happened to the centriole in each one of them? Fore claret-draw a picture describing each phenotype.

Quantitative analysis of the two types of defective HTCA ultrastructure in *Centlein*^{-/-} spermatids, i.e. the detached coupling apparatus from the spermatid nucleus and unassembled or destroyed coupling apparatus, has now been provided, respectively (Fig. R1). Apparently, the centriole(s) are present in *Centlein*^{-/-} spermatids, but whether or not their structures are normal remains unknown. We therefore feel awkward to draw a picture describing each phenotype.

Discussion

- Since CENTLEIN is a known centrosome protein there, needs to be a discussion of its role in the mice sperm that is thought not to have a centrosome (please see the paper "It Takes Two (Centrioles) to Tango"). This is important in light of the recent discovery that the human sperm centrioles are atypical (Fishman et al., 2018 - A novel atypical sperm centriole is functional during human fertilization. Nature Communications.).

Since CENTLEIN protein is undetectable at Step 15-16 spermatids and onwards, it is difficult for us to propose its functional role in the mature spermatozoa. To further confirm this result, we examined the expression of CENTLEIN in mature spermatozoa from caudal epididymis, and found CENTLEIN could not be detected in mature spermatozoa (Supplementary Fig. 3a). The immunofluorescent staining of CENTLEIN also showed the CENTLEIN was absent in mature spermatozoa (Supplementary Fig. 3b).

- "CENTLEIN, identified as the first centrosome protein, is essential for the integrity of the HTCA (Fig. 4)." What about this paper "Rat hd Mutation Reveals an Essential Role of Centrobin in Spermatid Head Shaping and Assembly of the Head-Tail Coupling Apparatus"

We sincerely apologize for the overstatement. This has now been rectified.
"Here, we show the centrosome protein, CENTLEIN, is essential for the integrity of the HTCA (Fig. 4f)."

Sincerely,
Tomer Avidopr-Reiss

Reference:

1. Frohnert C, Schweizer S, Hoyer-Fender S. SPAG4L/SPAG4L-2 are testis-specific SUN domain proteins restricted to the apical nuclear envelope of round spermatids facing the acrosome. *Mol Hum Reprod* **17**, 207-218 (2011).
2. Yassine S, Escoffier J, Abi Nahed R, Pierre V, Karaouzene T, Ray PF. Dynamics of Sun5 Localization during Spermatogenesis in Wild Type and Dpy19l2 Knock-Out Mice Indicates That Sun5 Is Not Involved in Acrosome Attachment to the Nuclear Envelope (vol 10, e0118698, 2015). *Plos One* **10**, (2015).
3. Shang Y, *et al.* Essential role for SUN5 in anchoring sperm head to the tail. *Elife* **6**, (2017).
4. Shang YL, *et al.* Mechanistic insights into acephalic spermatozoa syndrome-associated mutations in the human SUN5 gene. *Journal of Biological Chemistry* **293**, 2395-2407 (2018).
5. Hao H, Starr DA. SUN/KASH interactions facilitate force transmission across the nuclear envelope. *Nucleus* **10**, 73-80 (2019).
6. Karouzakis E, Gay RE, Gay S, Neidhart M. Increased recycling of polyamines is associated with global DNA hypomethylation in rheumatoid arthritis synovial fibroblasts. *Arthritis Rheum-Us* **64**, 1809-1817 (2012).
7. Zhu F, *et al.* Mutations in PMFBP1 Cause Acephalic Spermatozoa Syndrome. *Am J Hum Genet* **103**, 188-199 (2018).
8. Quintyne NJ, Gill SR, Eckley DM, Crego CL, Compton DA, Schroer TA. Dynactin is required for microtubule anchoring at centrosomes. *J Cell Biol* **147**, 321-334 (1999).
9. Quintyne NJ, Schroer TA. Distinct cell cycle-dependent roles for dynactin and dynein at centrosomes. *J Cell Biol* **159**, 245-254 (2002).
10. Guo J, *et al.* Nudel contributes to microtubule anchoring at the mother centriole and is involved in both dynein-dependent and -independent

- centrosomal protein assembly. *Molecular Biology of the Cell* **17**, 680-689 (2006).
11. Roosen-Runge EC. The process of spermatogenesis in mammals. *Biological reviews of the Cambridge Philosophical Society* **37**, 343-377 (1962).
 12. Hess RA, Renato de Franca L. Spermatogenesis and cycle of the seminiferous epithelium. *Advances in experimental medicine and biology* **636**, 1-15 (2008).
 13. Rathke C, Baarends WM, Awe S, Renkawitz-Pohl R. Chromatin dynamics during spermiogenesis. *Biochimica et biophysica acta* **1839**, 155-168 (2014).
 14. Bao J, Bedford MT. Epigenetic regulation of the histone-to-protamine transition during spermiogenesis. *Reproduction* **151**, R55-70 (2016).
 15. Govin J, Caron C, Lestrat C, Rousseaux S, Khochbin S. The role of histones in chromatin remodelling during mammalian spermiogenesis. *Eur J Biochem* **271**, 3459-3469 (2004).
 16. Fang G, Zhang D, Yin H, Zheng L, Bi X, Yuan L. Centlein mediates an interaction between C-Nap1 and Cep68 to maintain centrosome cohesion. *J Cell Sci* **127**, 1631-1639 (2014).
 17. Kim K, Lee S, Chang J, Rhee K. A novel function of CEP135 as a platform protein of C-NAP1 for its centriolar localization. *Experimental Cell Research* **314**, 3692-3700 (2008).
 18. Fishman EL, *et al.* A novel atypical sperm centriole is functional during human fertilization (vol 9, 2018). *Nature Communications* **9**, (2018).

Reviewers' Comments:

Reviewer #1:

Remarks to the Author:

The authors have successfully responded to the majority of the issues that I raised in my initial review of their manuscript. However, there are several major concerns that still need to be addressed before I feel comfortable recommending the publication of this work. These concerns are outlined (red text) in response to the authors' responses (blue text) to my original review (black text) in the PDF attached here.

Reviewer #2:

Remarks to the Author:

This manuscript is much improved and was interesting to read. I have two minor issues that can be fixed by the authors: (1) The Centlein localization data interpretation and (2) the status of DC and PC structure in the Centlein mutant (see below).

- Line 219-231: You label the DC and PC of Centlein mutant in Figure 1f, but I am not sure how you make this call – please show normal PC and DC in the mutant or indicate in the text that you are not able to identify normal PC and DC in the mutant.
- Fig 5C, d, e, and f as well as SF 4 as well as SF% and other florescent pictures - This is nice data - please mark the PC and DC based on CEP135 and CETN labeling in steps 9-16 spermatids.
- “On the other hand, some CENTLEIN and CEP135 signals showed to be closer to the sperm nucleus than CETN1/2 during the elongation and differentiation of the spermatid (Fig.5d and Supplementary Fig. 4), which indicated that, besides centrioles, CENTLEIN might also localize near to the implantation fossa region of the sperm nucleus” – I disagree - To me, CENTLEIN appear to localize between the PC and DC in steps 9-16 spermatids.
- It would be helpful if you can add a figure that model and summaries all the data on SUN5, PMFBP1, and SPATA6 and CENTLEIN role in linking the tail to the head.

Tomer Avidor-Reiss

REVIEWER COMMENTS

Reviewer #1 (Remarks to the Author):

Major issues.

1) **Previous Comment:** The authors need to better describe how they envision that SUN5 functions in the HTCA. SUN5 is a member of the SUN family of inner nuclear membrane proteins, yet the authors fail to mention this important fact anywhere in their manuscript. Nor do they mention that SUN proteins are core components of the linker of nucleoskeleton and cytoskeleton complex due to their ability to form a transluminal interaction with the KASH/nesprin family of outer nuclear membrane proteins. Based on this information, do the authors propose that the normally nucleoplasmic N-terminus of SUN5 resides on the outer nuclear membrane where it is able to interact with CENTLEIN? Alternatively, does SUN5 function to recruit a KASH/nesprin protein to the outer nuclear membrane of the sperm nucleus that is capable of interacting with CENTLEIN? The authors need to discuss these opposing models in their manuscript.

Previous Response: These are the very important and insightful comments brought forward by Reviewer #1, for which we are very grateful. It has been reported that the overexpressed SUN5 in somatic cells is localized at the inner nuclear membrane¹. However, few work has addressed the precise localization of SUN5 in sperm nuclear envelope, as the complicated localization of SUN5 in different studies^{1,2}. Previously, we found SUN5 was first expressed in the nuclear envelope and later migrated to the coupling apparatus of the sperm during sperm head elongation and differentiation ³. Furthermore, the SUN5C (coiled-coil domain and the SUN domain of SUN5) region of SUN5 does not interact with KASH5 LR⁴, indicating SUN5 might be distinct from the classical SUN family proteins. By using LC-MS analysis, we found SUN5 could directly bind to the type II heat shock protein 40 protein, DNAJB13, which might facilitate SUN5 protein folding to ensure the interaction between the implantation fossa and unknown protein in the HTCA ⁴.

Therefore, SUN5 might directly bind some cytoplasm proteins to achieve its physiological function during spermatogenesis. To further explore the underlying mechanism of SUN5 recruitment of CENTLEIN, we firstly mapped the CENTLEIN binding region in SUN5 according Reviewer #1's suggestion, and found the SUN domain of SUN5 was sufficient to bind CENTLEIN (Fig. 6c). In addition, the GST-pulldown experiments showed the SUN domain of SUN5 could directly interact with the 971–1396aa region of CENTLEIN (Fig. 6e). The immunofluorescent staining also showed CENTLEIN could partially colocalize with SUN5 in the HTCA (Fig. 5e). Thus, SUN5 might directly recruit CENTLEIN through its SUN domain during spermatogenesis. Many thanks for helping us clarify this important issue, and we have introduced the cellular functions of SUN5 in the introduction part and discussed the potential model of SUN5 binding CENTLEIN in the discussion as below.

New Comment:-I appreciate the authors making the effort to address this important issue.

However, I am rather concerned about the physiological relevance of the reported SUN5 SUN domain-CENTLEIN interaction and this concern also applies to the previously reported SUN5-DNAJB13 interaction. My concern regarding these interactions arises as a function of topology. To date, I am unaware of any SUN protein that does not contain a C-terminal SUN domain-containing luminal domain. In contrast, both CENTLEIN and DNAJB13 are both cytosolic proteins. Therefore, if the SUN domain of SUN5 were to interact with either CENTLEIN or DNAJB13 the normally luminal SUN5 C-terminus would need to be present within the cytoplasm. Consequently, I strongly suggest that the authors determine the topology of SUN5 in the sperm nuclear envelope. One strategy would be to compare the immunofluorescence staining of sperm permeabilized with digitonin or Triton-X100 using an anti-SUN5 antibody that recognizes the C-terminus of the protein. It may very well be that SUN5 can interact with CENTLEIN; however, I would anticipate that this interaction would be indirect in nature.

R: This is an excellent suggestion. To determine the topology of SUN5 in the sperm nuclear envelope, we performed the suggested experiments as follows.

(1) The isolated round spermatids permeabilized with digitonin or Triton-X100 were stained with different combination of antibodies against SUN5, the outer nuclear membrane (ONM) marker SYNE1, the inner nuclear membrane (INM) LAP2 or the nuclear lamina marker LAMIN B1^{5,6,7}. Of note, the ONM marker SYNE1 and a marker for INM LAP2 and LAMIN B1 were selected because all of three were present in the mouse spermatids^{5,6,7}.

Given that digitonin permeabilization selectively disrupts the plasma membrane leaving the NE membranes intact⁸ and Triton X-100 permeabilizes all membranes, antibodies to both LAP2 and LAMIN B1 stained the NE only in Triton X-100-permeabilized spermatids (Supplementary Fig.7a, b), whereas the antibody to the ONM marker SYNE1 labeled the spermatid NE after selective digitonin permeabilization (Supplementary Fig.7c). Of importance, we noticed that the SUN5-labelling pattern in digitonin-permeabilized spermatids using the mouse anti-SUN5 antibody against the SUN5 SUN domain was indistinguishable from that in Triton X-100-treated spermatids (Supplementary Fig.7a, c), in that both exhibited an NE staining, indicating that the SUN5 SUN domain being detected by the antibody is exposed to the spermatid cytosol.

(2) The round spermatids with a purity of more than 85%⁹ were subjected to the assay of in situ proteinase K digestions¹⁰. As previously described¹⁰, treatment with low concentrations of digitonin permeabilizes the plasma membrane but not the nuclear membranes. Subsequent proteinase K digestion then degrades cytoplasmic and nuclear proteins, whereas ER luminal and perinuclear space (PNS) proteins were protected. Treatment with Triton X-100 and proteinase K lead to the digestion of all those proteins, whereas incubation of cells with proteinase K without prior membrane permeabilization prevents the degradation. We reconstituted the assay in round spermatids, as ascertained by cytoplasmic Tubulin and nuclear LAMIN B1 (Supplementary Fig. 7e).

When probed with the verified mouse anti-SUN5 antibody against the SUN5 SUN domain (Supplementary Fig. 7d), a band of the expected size for SUN5 was seen in protein extracts from the round spermatids (Supplementary Fig. 7e). Triton X-100 permeabilization in conjunction with proteinase K digestion resulted in degradation of cytoplasmic Tubulin, nuclear LAMIN B1 and SUN5 (Supplementary Fig. 7e). When round spermatids permeabilized with digitonin followed by proteinase K digestion, Western blot analysis revealed that the level of SUN5 probed with the mouse anti-SUN5 antibody against the SUN5 SUN domain diminished (Supplementary Fig. 7e), which showed a similar phenomenon with the cytoplasmic Tubulin and nuclear LAMIN B1. These results further support the SUN domain of SUN5 exposed to the cytoplasm in the spermatids.

By performing in situ proteinase K digestion assays and digitonin experiments, we demonstrated the SUN5 SUN domain could be present within the cytoplasm of the spermatids, in turn, rendering the SUN5-CENTLEIN interaction in the spermatids.

Previous Response: *“SUN5 is a transmembrane protein located on the nuclear envelope, which could interact with the coupling apparatus-related protein DnaJ heat shock protein family (Hsp40) member B13 (DNAJB13) to facilitate SUN5 protein folding to ensure the interaction between the implantation fossa and an unknown protein in the HTCA.”*

New Comment: -As discussed above, I would strongly recommend that the authors reconsider the physiological relevance of the SUN5 SUN domain-DNAJB13 interaction due to topology issues. I would also recommend that the authors change the word “on” found after “located” to “in” in the text above.

R: We thank the reviewer for the comments.

Because SUN5 protein has a transmembrane domain, this domain definitely

localizes in the nuclear envelope, and we have changed the word “on” found after “located” to “in” in the text.

Previous Response: *“SUN5 belongs to the SUN domain protein in the inner nuclear membrane, which could fasten the linkage between the nuclear lamina and cytoskeleton through the SUN-KASH interfaces^{1, 11}. Although the overexpressed SUN5 in somatic cells is localized at the inner nuclear membrane¹, few work has addressed the precise localization of SUN5 in sperm nuclear envelope, as the complicated localization of SUN5 in different studies^{1, 2}. Previously, we found the SUN5C (coiled-coil domain and the SUN domain of SUN5) region of SUN5 does not interact with KASH5 LR4, indicating SUN5 might be distinct from the classical SUN domain proteins. By using LC-MS analysis, we found SUN5 could directly bind to the DNAJB13 to facilitate SUN5 protein folding, which is required for the integrity of the HTCA⁴. Here, we found the SUN domain of SUN5 could directly interact with the 971–1396aa region of CENTLEIN (Fig. 6e). The immunofluorescent staining showed CENTLEIN could partially colocalize with SUN5 in the HTCA (Fig. 5e), and absence of SUN5 might perturb the HTCA localization of CENTLEIN (Fig. 7 i, j). Thus, SUN5 might directly recruit CENTLEIN to the HTCA through its SUN domain.”*

New Comment: -As discussed above, I would strongly recommend that the authors reconsider the physiological relevance of the SUN5 SUN domain-DNAJB13 and –CENTLEIN interactions due to topology issues.

R: We reconsider the physiological relevance of the SUN5 SUN domain-DNAJB13 and -CENTLEIN interactions.

(1) DNAJB13 facilitates SUN5 protein folding at the HTAC of mouse spermatids and spermatozoa via the SUN5 SUN domain-DNAJB13 interaction⁴.

(2) SUN5 SUN domain-CENTLEIN interaction is involved in the HTCA assembly and integration of mouse sperm head to the tail.

(3) SUN5 and DNAJB13 are encoded by two disease-causing genes^{12, 13}, while the involvement of the *CENTLEIN* gene in human disease is eagerly awaited.

New Comment: I would also recommend the following changes in the text presented above:

-Change "SUN5 belongs to the SUN domain protein in the inner nuclear membrane" to "SUN5 belongs to the SUN domain-containing family of inner nuclear membrane proteins".

R: Thank you for your suggestion, we have revised it as "SUN5 belongs to the SUN domain-containing family of nuclear membrane proteins".

-Change "which could fasten the linkage between" to "which physically couple".

R: We have revised this part.

-Change "the SUN-KASH interfaces" to "the assembly of linker of nucleoskeleton and cytoskeleton (LINC) complexes".

R: We have revised it.

-Define "LC-MS".

R: We have defined it as "Liquid Chromatography Mass Spectrometry (LC-MS)".

-Change "The immunofluorescent staining" to "The immunofluorescence staining".

R: Sorry for this mistake. We have revised it.

-Change "could partially colocalize with SUN5 in the HTCA" to "partially colocalizes with SUN5 in the HTCA".

R: We have revised it.

Change "absence of SUN5 might perturb" to "the absence of SUN5 perturbs".

R: We have revised this part.

Previous Response: *“PMFBP1 was firstly identified as a polyamine modulating factor 1 (PMF1) binding protein, which could enhance the catabolism and recycling of polyamines⁶. During spermatogenesis, PMFBP1 is specifically expressed in adult testis and predominantly located in the HTCA.”*

New Comment: OK. Please change “adult testis” to “adult testes”.

R: We thank the reviewer and have made the change accordingly.

Previous Comment: 3) Continuing along the theme of mechanism of SUN5 function described above, the authors really need to map the CENTLEIN binding site in SUN5. They also should map the SUN5 and PMFBP1 binding sites on CENTLEIN. Moreover, they need to test if the SUN5-CENTLEIN interaction were direct or not using recombinant proteins. Despite what the authors state in line 95 of the Introduction, the ability of two proteins to be co-immunoprecipitated does not prove that they are able to directly interact with each other. While I understand that the pandemic makes it difficult to perform additional experiments at this time, I strongly feel that a deeper mechanistic understanding of the CENTLEIN/SUN5/PMFBP1 interaction is needed for this manuscript to be suitable for publication at Nat Commun.

Previous Response: We thank the reviewer for the constructive comments.

(1) Deletion analysis of CENTLEIN showed that aa 971-1396 was sufficient to bind SUN5 (Fig. 6a), and 601-1396aa was required for its binding to PMFBP1 (Fig. 6b). Two portions of PMFBP1 encompassing residues 1-282 and 750-1023 were necessary for its binding to CENTLEIN (Fig. 6d), while SUN domain of SUN5 was sufficient to bind CENTLEIN (Fig. 6c).

New Comment: As discussed above, I question the physiological relevance of the SUN5 SUN domain-CENTLEIN interaction based on potentially topology issues. If the SUN domain of SUN5 were to reside within the perinuclear space of the nuclear envelope, it would not be present within the cytoplasm where CENTLEIN resides.

While the GFP-SUN5 193-373aa construct was able to co-immunoprecipitate myc-CENTLEIN in Figure 6C, it would most likely not be properly targeted to the lumen of the endoplasmic reticulum nor the contiguous perinuclear space of the nuclear envelope due to the absence of a signal sequence or hydrophobic stop-transfer sequence.

R: Our experimental results do not support the speculation of this reviewer, as shown in Supplementary Fig. 7, anti-SUN5 SUN domain antibodies did not display any different staining pattern in the cells permeabilized with digitonin or Triton X-100, suggesting that the SUN5 SUN domain being detected by the antibody is exposed to the spermatid cytosol, in turn, rendering the SUN5-CENTLEIN interaction in the mouse spermatids.

Previous Response: (2) An *in vitro* direct binding assay was subsequently conducted. GST-SUN domain of SUN5 readily pulldowned CENTLEIN 971-1396aa (Fig. 6e), and recombinant PMFBP1 750-1023aa could directly interact with CENTLEIN 601-970aa (Fig. 6f).

New Comment: As discussed above, I question the physiological relevance of the SUN5 SUN domain-CENTLEIN interaction based on potentially topology issues.

R: We performed the suggested experiments explicitly. However, our data presented in Supplementary Fig. 7 supported the SUN5 SUN domain could be present within the cytoplasm, thus rendering the SUN5-CENTLEIN interaction in the mouse spermatids.

Previous Response: (3) We have now shown that CENTLEIN mediates an interaction between SUN5 and PMFBP1 (Fig. 6g). This evidence, along with a lack of interaction between SUN5 and PMFBP1,¹⁴ suggests that SUN5 complexes with PMFBP1 through the independent direct interactions of both proteins with CENTLEIN (Fig.6).

New Comment: As discussed above, I question the physiological relevance of the SUN5 SUN domain-CENTLEIN interaction based on potentially topology issues.

R: The response can be found for the same question before.

Previous Comment: 4) The authors fail to mention the source of the cDNA constructs that encode the EGFP-tagged centrosome proteins used in Figure 1. Are these previously described constructs? Are they functional? If they are not, the authors should provide representative images of the subcellular localization of these EGFP-tagged centrosome proteins in the HEK293T cells to demonstrate that these constructs are properly localized to the centrosome.

Previous Response: We sincerely apologize for not mentioning the source of the cDNA constructs that encode the EGFP-tagged centrosome proteins used in Figure 1. Now, we have provided the information of these cDNA constructs in Methods. Centrosome localization of these EGFP-tagged proteins are now presented in Supplemental Figure 1. P50 is a subunit of the dynactin complex localized at the centrosome^{15,16,17}, and its overexpression has been used to study dynactin-dynein-related centrosome proteins, albeit without clear centrosome localization^{15,16,17}.

New Comment: Thank you for the clarification. What exactly do you mean by “overexpression has been used to study dynactin-dynein-related centrosome proteins”? What has been studied about these proteins using p50 over expression?

R: We thank the reviewer for the question that promotes us to have an in-depth search for its original use for centrosome studies.

(1) Dynactin contains a total of 20 individual polypeptide subunits that are encoded by 11 different genes^{15,18}. Dynamitin (p50) is a major structural component of the dynactin shoulder¹⁹, and its overexpression causes the shoulder to be displaced

from the rest of the dynactin structure, rendering defocusing of the radial microtubule array and a redistribution of the pericentriolar proteins, such as γ -Tubulin, Dynactin, PCM-1, Centrin, and Ninein^{15, 16, 20}.

(2) A number of proteins transported to the centrosome depend on the dynein–dynactin complex. To test whether the dynein–dynactin complex is necessary for centrosomal delivery of a protein of interest, overexpressed exogenous Dynamitin (p50), a well-characterized inhibitor of the dynein–dynactin complex, is often used^{17, 20, 21, 22, 23, 24}.

(3) We have now realized that application of p50 overexpression is inappropriate in the present study. We therefore choose to remove the GFP-p50 from Fig. 1a and Supplementary Fig. 1. We deeply apologize for the improper use.

Previous Comment: d. What are the big black bands found in the FLAG blot shown in Figure 1G? Are they IgG light or heavy chains? The amount of over-saturation of these big black bands makes it difficult to see the immunoprecipitated FLAG-SPATA6 band.

Previous Response: Yes, they are IgG heavy chain(s).

New Comment: OK. This should be indicated in the appropriate blots shown in the Supplemental Figures.

R: Yes, this has been indicated in the Supplemental Figure 8.

Previous Comment: b. The authors need to provide some quantitative analysis of the TEM results shown in Figure 4C.

Previous Response: As a single cell could be cut into different sections, it might be captured repeatedly in TEM experiment, thus it was difficult to quantify the broken sperm head-tail junction by TEM. We have collected all TEM sections and tried to quantify the two types of defective HTCA ultrastructure in *Centlein*^{-/-} spermatids, i.e. the detached coupling apparatus from the spermatid nucleus and unassembled or destroyed coupling apparatus, has now been provided, respectively (Fig. R1).

New Comment: Fair enough. Perhaps Figure R1 could be made into a supplemental figure and included in the manuscript?

R: The figure has been made into a supplemental figure labeled as Supplemental Figure 3.

Previous Comment: b. The authors need to provide some quantitative analysis of the immunofluorescence results shown in Figures 5C-D. For example, what is the extent of the co-localization of CEP135 and CENTLEIN over the course of spermatogenesis? What is the extent of the co-localization of SUN5 and CENTLEIN or PMFBP1 and CENTLEIN?

Previous Response: This is a very important point brought forward by the referee.

(1) Using Super-Resolution microscopy and an antibody against CETN1/2, a standard marker for centrioles in mammalian spermatids, we found that CENTLEIN localized to both distal and proximal centrioles of round, elongating, and early elongated spermatids (Fig. 5d).

New Comment: How exactly did the authors use Imaris to determine the “% pixel overlap”? It may be better to report the Pearson’s correlation coefficients and some line scans.

R: We sincerely apologize for not presenting how to use Imaris to determine the “% pixel overlap”. We have now provided how to use Imaris to determine the “% pixel overlap” below and in **Methods**.

Co-localization analysis was performed utilizing Imaris 9.0.2 software. Using Imaris surface module and background subtraction function, we calculated the volume of green and red channels, respectively. Using Imaris coloc module, we set a threshold, and then calculated the volume of the new extracted co-localization channel. The level of co-localization was expressed as the percentage of the co-localization volume to the volume of green and red channels, respectively.

We fully agreed. The Pearson’s correlation coefficients (Imaris software) and line-scan analysis (ZEN and Leica Application Suite X software) have now been added in Fig. 5c.

The image analysis in this manuscript were described in the Methods section as below:

“Co-localization analysis were performed utilizing IMARIS software. The level of co-localization in the three-dimensional volume was measured as percent of volume of the channel co-localized. Using IMARIS surface module and background subtraction function, we calculated the volume of green and red channels, respectively. Using IMARIS coloc module, we set a threshold and calculated the volume of a new extracted co-localization channel. The level of co-localization was expressed as the percentage of the co-localization volume to the volume of the green or red channel respectively. A second measure of the intensity of co-localization between two signals was obtained by calculating the correlation between the intensities of the co-localized three-dimensional pixels (Pearson’s correlation coefficient). The extent of colocalization of two labels was measured using the Coloc module. The intensity threshold in both channels was automatically determined. The Pearson’s correlation coefficient lies between +1 and -1, with positive values indicating a direct correlation and values near 0 indicating no correlation. Relative pixel intensities of fluorescence were analyzed by

line-scan analysis using ZEN software or Leica Application Suite X software. The position of line-scan is indicated by a white line on the merged image. Using Profile module and line scan function, we measured intensity of two fluorescence signals (red and green) across the white line.”

Previous Response: (4) Co-localization between SUN5 and CENTLEIN, as well as PMFBP1 and CENTLEIN occurs during Step 10-14. The extent of the co-localization of SUN5 and CENTLEIN or PMFBP1 and CENTLEIN has been provided in Fig. 5e and 5f.

New Comment: See my comment above regarding “% overlap”.

R: We fully agreed. The details of how to use Imaris to determine the “% pixel overlap” and the Pearson’s correlation coefficients have now been provided above. Of note, the line-scan presented in Fig. 5e, f. was analyzed with Leica Application Suite X software.

Previous Comment: c. Why does SUN5 show such a punctate localization in the sperm nuclear envelope? Is this a result of the particular optical section shown in Figure 5D? In addition, what are the clumps of PMFBP1 that do not co-localize with CENTLEIN in Figure 5E? How specific are the antibodies for SUN5 and PMFBP1? Do the authors see the same subcellular localization of SUN5 and PMFBP1 with other antibodies for these proteins?

Previous Response: We thank the reviewer for questioning the specificity of the anti-SUN5 and anti- PMFBP1 antibodies used in the previous manuscript. The new batches of the anti-SUN5 and anti-PMFBP1 antibodies have been employed in the revised manuscript (Fig. 5e, f).

New Comment: What do the authors mean by “the new batches of anti-SUN5 and anti-PMFBP1 antibodies”? Are these antibodies more specific for their respective

antigens? If so, how was this determined? In addition, the authors need to describe how the “mouse SUN5 antibody for immunofluorescence (1:100)” was generated somewhere in the Materials and Methods.

R: We thank the reviewer for spotting this confusion.

(1) Because both anti-SUN5 and anti-PMFBP1 antibodies employed in NCOMMS-20-11518 were used up, we have to purchase a new batch of those two antibodies.

(2) The rabbit anti-SUN5 polyclonal antibody (17495-1-AP) against the full-length of human SUN5 (1-379aa).

The rabbit anti- PMFBP1 polyclonal antibody (17061-1-AP) against residues 673-1022 of the human PMFBP1.

(3) The mouse SUN5 antibody for immunofluorescence (1:100) is an in-house-generated antibody against the SUN5 SUN domain.

(4) Upon arrival of these antibodies, we immediately optimize their conditions for immunofluorescence. Of note, use of freshly prepared samples/slides is critical for getting specific signals.

Previous Comment: c. The images of SUN5 provided in Figure 6B make me seriously question the specificity of the anti-SUN5 antibody used in this work. Given this concern, I find it difficult to be able to agree with the authors’ conclusion that “Ablation of Centlein impairs the localization of PMFBP1 to the coupling apparatus.

Previous Response: The SUN5-specific antibody has been used in the revised manuscript (Fig. 7d, g).

New Comment: Which “SUN5-specific antibody” was used? See comments above.

R: We thank the reviewer for spotting this confusion.

“SUN5-specific antibody” means the antibody specifically recognizing SUN5 (Supplementary Fig. 7d). In order to eliminate / minimize non-specific staining, we have optimized the experimental conditions for immunofluorescence.

Previous Comment: 3) In the Epididymal Sperm Count section of the Methods, the authors need to provide more information regarding the microscopes used to generate the data presented in this work. Specifically, they need to report which objectives (magnification, correction level, N.A.), detectors, light sources, software, excitation filters, and emission filters were used.

Previous Response: The detail information regarding the microscopes has been provided in the revised manuscript.

New Comment: What about the microscope's camera, light source, and software?

R: (1) Light source of the microscope: LED: 3W 3200k.

(2) The microscope equipped with neither camera nor software.

(3) Hemocytometer counting was used for the Epididymal Sperm Count.

The whole procedure, often used for mouse epididymal sperm count, is as follows:

"The caudal epididymis was dissected from 8-week-old mice. Spermatozoa were squeezed out from the caudal epididymis and incubated for 30 min at 37°C under 5% CO₂. The incubated sperm medium was then diluted 1:10. A cover slip was placed on the hemocytometer before a drop with 10µl of diluted caudal epididymal sperm solution was loaded under the cover slip. The hemocytometer was placed under the Primo Star microscope (Zeiss) and viewed under ×400 magnification. The microscope equipped with neither camera nor software, and the light source of the microscope is LED: 3W 3200k. Sperm count was done by counting 4×4 squares (horizontally or vertically) using the hemocytometer and calculated using the formula below: Sperm count = total no. of sperm in 5 squares × 50,000 × 1000 × dilution multiple (cells/L). Counting was only done for sperm tails that was found within the square areas. The total number of sperm was counted and the mean was calculated from three counts. The 6 independent experiments were performed. The data were then analyzed with Graphpad Prism7."

Reviewer #2 (Remarks to the Author):

This manuscript is much improved and was interesting to read. I have two minor issues that can be fixed by the authors: (1) The Centlein localization data interpretation and (2) the status of DC and PC structure in the Centlein mutant (see below).

R: We thank Prof. Avidopr-Reiss for his positive qualifications of the work: “This manuscript is much improved and was interesting to read.”

- Line 219-231: You label the DC and PC of Centlein mutant in Figure 1f, but I am not sure how you make this call – please show normal PC and DC in the mutant or indicate in the text that you are not able to identify normal PC and DC in the mutant.

R: By performing dozens of new TEM sections, we are now able to label PC and DC in *Centlein*^{-/-} spermatids (Fig. 4).

- Fig 5C, d, e, and f as well as SF 4 as well as SF% and other florescent pictures - This is nice data - please mark the PC and DC based on CEP135 and CETN labeling in steps 9-16 spermatids.

R: We thank Prof. Avidopr-Reiss for spotting this missing. PC and DC have now been marked (Fig. 5d and Supplementary Fig. 5).

- “On the other hand, some CENTLEIN and CEP135 signals showed to be closer to the sperm nucleus than CETN1/2 during the elongation and differentiation of the spermatid (Fig.5d and Supplementary Fig. 4), which indicated that, besides centrioles, CENTLEIN might also localize near to the implantation fossa region of the sperm nucleus” – I disagree - To me, CENTLEIN appear to localize between the PC and DC in steps 9-16 spermatids.

R: We agree with this comment, and the corresponding statement has been removed.

- It would be helpful if you can add a figure that model and summaries all the data on SUN5, PMFBP1, and SPATA6 and CENTLEIN role in linking the tail to the head.

R: Yes, the model has now been provided, labeled as Figure 8.

Tomer Avidor-Reiss

Reference:

1. Frohnert C, Schweizer S, Hoyer-Fender S. SPAG4L/SPAG4L-2 are testis-specific SUN domain proteins restricted to the apical nuclear envelope of round spermatids facing the acrosome. *Mol Hum Reprod* **17**, 207-218 (2011).
2. Yassine S, Escoffier J, Abi Nahed R, Pierre V, Karaouzene T, Ray PF. Dynamics of Sun5 Localization during Spermatogenesis in Wild Type and Dpy19l2 Knock-Out Mice Indicates That Sun5 Is Not Involved in Acrosome Attachment to the Nuclear Envelope (vol 10, e0118698, 2015). *Plos One* **10**, (2015).
3. Shang Y, *et al.* Essential role for SUN5 in anchoring sperm head to the tail. *Elife* **6**, (2017).
4. Shang YL, *et al.* Mechanistic insights into acephalic spermatozoa syndrome-associated mutations in the human SUN5 gene. *Journal of Biological Chemistry* **293**, 2395-2407 (2018).
5. Schutz W, Alsheimer M, Ollinger R, Benavente R. Nuclear envelope remodeling during mouse spermiogenesis: postmeiotic expression and redistribution of germline lamin B3. *Exp Cell Res* **307**, 285-291 (2005).

6. Alsheimer M, Fecher E, Benavente R. Nuclear envelope remodelling during rat spermiogenesis: distribution and expression pattern of LAP2/thymopoietins. *J Cell Sci* **111 (Pt 15)**, 2227-2234 (1998).
7. Gob E, Schmitt J, Benavente R, Alsheimer M. Mammalian sperm head formation involves different polarization of two novel LINC complexes. *PLoS One* **5**, e12072 (2010).
8. Adam SA, Marr RS, Gerace L. Nuclear protein import in permeabilized mammalian cells requires soluble cytoplasmic factors. *J Cell Biol* **111**, 807-816 (1990).
9. Bellve AR, Cavicchia JC, Millette CF, O'Brien DA, Bhatnagar YM, Dym M. Spermatogenic cells of the prepuberal mouse. Isolation and morphological characterization. *J Cell Biol* **74**, 68-85 (1977).
10. Liu Q, *et al.* Functional association of Sun1 with nuclear pore complexes. *J Cell Biol* **178**, 785-798 (2007).
11. Hao H, Starr DA. SUN/KASH interactions facilitate force transmission across the nuclear envelope. *Nucleus* **10**, 73-80 (2019).
12. Zhu F, *et al.* Biallelic SUN5 Mutations Cause Autosomal-Recessive Acephalic Spermatozoa Syndrome. *Am J Hum Genet* **99**, 942-949 (2016).
13. El Khouri E, *et al.* Mutations in DNAJB13, Encoding an HSP40 Family Member, Cause Primary Ciliary Dyskinesia and Male Infertility. *Am J Hum Genet* **99**, 489-500 (2016).
14. Zhu F, *et al.* Mutations in PMFBP1 Cause Acephalic Spermatozoa Syndrome. *Am J Hum Genet* **103**, 188-199 (2018).
15. Quintyne NJ, Gill SR, Eckley DM, Crego CL, Compton DA, Schroer TA. Dynactin is required for microtubule anchoring at centrosomes. *J Cell Biol* **147**, 321-334 (1999).

16. Quintyne NJ, Schroer TA. Distinct cell cycle-dependent roles for dynactin and dynein at centrosomes. *J Cell Biol* **159**, 245-254 (2002).
17. Guo J, *et al.* Nudel contributes to microtubule anchoring at the mother centriole and is involved in both dynein-dependent and -independent centrosomal protein assembly. *Mol Biol Cell* **17**, 680-689 (2006).
18. Lennarz WJ, Lane MD. *Encyclopedia of biological chemistry*, Second edition. edn. Elsevier (2013).
19. Urnavicius L, *et al.* The structure of the dynactin complex and its interaction with dynein. *Science* **347**, 1441-1446 (2015).
20. Dammermann A, Merdes A. Assembly of centrosomal proteins and microtubule organization depends on PCM-1. *J Cell Biol* **159**, 255-266 (2002).
21. Ye X, Zeng H, Ning G, Reiter JF, Liu A. C2cd3 is critical for centriolar distal appendage assembly and ciliary vesicle docking in mammals. *Proc Natl Acad Sci U S A* **111**, 2164-2169 (2014).
22. Kodani A, Tonthat V, Wu B, Sutterlin C. Par6 alpha interacts with the dynactin subunit p150 Glued and is a critical regulator of centrosomal protein recruitment. *Mol Biol Cell* **21**, 3376-3385 (2010).
23. Lu F, Lan R, Zhang H, Jiang Q, Zhang C. Geminin is partially localized to the centrosome and plays a role in proper centrosome duplication. *Biol Cell* **101**, 273-285 (2009).
24. Wiens CJ, *et al.* Bardet-Biedl syndrome-associated small GTPase ARL6 (BBS3) functions at or near the ciliary gate and modulates Wnt signaling. *J Biol Chem* **285**, 16218-16230 (2010).

Reviewers' Comments:

Reviewer #1:

Remarks to the Author:

Overall, I feel that the authors have successfully addressed the majority of the issues that I had raised in my previous review of their initially submitted manuscript. However, I still have some major reservations regarding the specificity of their anti-SUN5 SUN domain antibodies, which were used to determine the topology of SUN5 in testicular germ cell nuclear envelopes. For example, the full Western blots provided for the authors' mouse anti-SUN5 SUN domain antibody shown in Fig. S9 for Fig. S7d (i.e. their digitonin-based SUN5 topology experiments) reveal the existence of several protein bands in addition to what the authors indicate is the SUN5-specific band. Given the presence of these additional immunoreactive bands, I feel that the authors really need to provide their readers with immunofluorescence images of testicular germ cells and sperm isolated from SUN5^{+/+} and SUN5^{-/-} mice so that we can determine for ourselves how specific this antibody is for SUN5 in immunofluorescence experiments. This is especially important for the digitonin-based SUN5 topology experiments, which were shown in Fig. S7. The authors show perform a similar experiment with their rabbit anti-SUN5 SUN domain antibody so that its specificity can also be evaluated. Without this important information, I still find it difficult to understand how the presumably normally luminal SUN domain of SUN5 may be found in the cytoplasm in sperm.

Reviewer #1 (Remarks to the Author):

Overall, I feel that the authors have successfully addressed the majority of the issues that I had raised in my previous review of their initially submitted manuscript. However, I still have some major reservations regarding the specificity of their anti-SUN5 SUN domain antibodies, which were used to determine the topology of SUN5 in testicular germ cell nuclear envelopes. For example, the full Western blots provided for the authors' mouse anti-SUN5 SUN domain antibody shown in Fig. S9 for Fig. S7d (i.e. their digitonin-based SUN5 topology experiments) reveal the existence of several protein bands in addition to what the authors indicate is the SUN5-specific band. Given the presence of these additional immunoreactive bands, I feel that the authors really need to provide their readers with immunofluorescence images of testicular germ cells and sperm isolated from SUN5+/+ and SUN5-/- mice so that we can determine for ourselves how specific this antibody is for SUN5 in immunofluorescence experiments. This is especially important for the digitonin-based SUN5 topology experiments, which were shown in Fig. S7. The authors show perform a similar experiment with their rabbit anti-SUN5 SUN domain antibody so that its specificity can also be evaluated. Without this important information, I still find it difficult to understand how the presumably normally luminal SUN domain of SUN5 may be found in the cytoplasm in sperm.

R: We thank the reviewer for giving us the chance to clarify the confusion in Supplementary Fig.7.

(1) We apologize for not presenting our data in a clearer way. The rabbit anti-SUN5 polyclonal antibody displayed in Supplementary Fig.7b is not the antibody against the SUN5 SUN domain, but against the recombinant full-length SUN5 protein (17495-1-AP, ProteinTech). The specificity of the antibody was verified via both Western blotting and immunofluorescence experiments (Fig. R1) ^{1, 2}. To avoid further confusion, the rabbit anti-SUN5 polyclonal antibody

against the recombinant full-length SUN5 protein is no longer present in Supplementary Fig. 7.

Figure R1. The specificity of the antibody against the full-length SUN5 protein (17495-1-AP, ProteinTech) was verified via both Western blotting and immunofluorescence experiments in the previous works.

(2) As suggested with the mouse anti-SUN5 SUN domain antibody, immunofluorescence images of testicular germ cells and sperm isolated from *Sun5*^{+/+} and *Sun5*^{-/-} mice have now been provided in Supplementary Fig.7a.

Reference:

1. Shang YL, *et al.* Mechanistic insights into acephalic spermatozoa syndrome-associated mutations in the human SUN5 gene. *Journal of Biological Chemistry* **293**, 2395-2407 (2018).

2. Shang Y, *et al.* Essential role for SUN5 in anchoring sperm head to the tail. *Elife* **6**, (2017).

Reviewers' Comments:

Reviewer #1:

Remarks to the Author:

The authors have now successfully addressed my concerns. I approve this manuscript for publication.

REVIEWERS' COMMENTS

Reviewer #1 (Remarks to the Author):

The authors have now successfully addressed my concerns. I approve this manuscript for publication.

Response:

We thank the reviewer for his/her comments.